# *Caenorhabditis elegans* SEL-5/AAK1 regulates cell migration and cell outgrowth independently of its kinase activity

**Filip Knop[1], Apolena Zounarová[1], Vojtěch Šabata[1], Teije Corneel Middelkoop[2], Marie Macůrková[1]***

[1]Department of Cell Biology, Faculty of Science, Charles University, Prague, Czech Republic; [2]Institute of Molecular Genetics, Czech Academy of Sciences, Prague, Czech Republic

**Abstract** During *Caenorhabditis elegans* development, multiple cells migrate long distances or extend processes to reach their final position and/or attain proper shape. The Wnt signalling pathway stands out as one of the major coordinators of cell migration or cell outgrowth along the anterior-posterior body axis. The outcome of Wnt signalling is fine-tuned by various mechanisms including endocytosis. In this study, we show that SEL-5, the *C. elegans* orthologue of mammalian AP2-associated kinase AAK1, acts together with the retromer complex as a positive regulator of EGL-20/Wnt signalling during the migration of QL neuroblast daughter cells. At the same time, SEL-5 in cooperation with the retromer complex is also required during excretory canal cell outgrowth. Importantly, SEL-5 kinase activity is not required for its role in neuronal migration or excretory cell outgrowth, and neither of these processes is dependent on DPY-23/AP2M1 phosphorylation. We further establish that the Wnt proteins CWN-1 and CWN-2, together with the Frizzled receptor CFZ-2, positively regulate excretory cell outgrowth, while LIN-44/Wnt and LIN-17/Frizzled together generate a stop signal inhibiting its extension.

**\*For correspondence:**
silhankm@natur.cuni.cz

**Competing interest:** The authors declare that no competing interests exist.

## Editor's evaluation

This important study defines developmental roles for a protein kinase involved in endocytosis and reports a surprising finding that the kinase catalytic activity is unnecessary. The evidence supporting the claims of the authors is solid. As this kinase was previously suggested to be a target of drug design efforts, this work will be of broad interest to cell biologists and biochemists.

## Introduction

During the development of *Caenorhabditis elegans*, several cells, both neuronal and non-neuronal, migrate in a well-defined and invariant manner along the anterior-posterior (A-P) body axis. Other cells do not migrate but rather extend their processes along the same axis. In both instances, the directionality of migration and growth is controlled by a set of guidance cues provided by the surrounding tissue and received by the corresponding signalling apparatus in the migrating or growing cell (*Silhankova and Korswagen, 2007*; *Sundaram and Buechner, 2016*; *Sherwood and Plastino, 2018*; *Hutter, 2019*). The gradients of Wnt proteins and a panel of their receptors are the key determinants of A-P guidance in *C. elegans*. Wnts can act either as attractants, repellents, or permissive cues for a given cell and the cell response depends on a given combination of Wnts, Frizzleds, and other components

of the Wnt pathway expressed in and around that particular cell (*Silhankova and Korswagen, 2007*; *Ackley, 2014*; *Middelkoop and Korswagen, 2014*; *Rella et al., 2016*).

Endocytosis is an important regulatory mechanism for Wnt signalling applied in both Wnt-producing and Wnt-receiving cells. On the way out of the producing cell, Wnts are accompanied by a Wnt sorting receptor Wntless (WLS) (*Bänziger et al., 2006*; *Bartscherer et al., 2006*; *Goodman et al., 2006*). Once WLS is relieved of its cargo at the plasma membrane, it is internalized by clathrin- and adaptor protein complex 2 (AP2)-dependent endocytosis (*Pan et al., 2008*; *Port et al., 2008*; *Yang et al., 2008*). Internalized WLS is subsequently recycled by a multisubunit retromer complex to the Golgi apparatus (*Belenkaya et al., 2008*; *Franch-Marro et al., 2008*; *Pan et al., 2008*; *Port et al., 2008*; *Yang et al., 2008*) and further to the endoplasmic reticulum (*Yu et al., 2014*). Disruption of the AP2 or clathrin function results in decreased Wnt secretion due to improper WLS recycling (*Pan et al., 2008*; *Port et al., 2008*; *Yang et al., 2008*). In Wnt-receiving cells, the activity of the Wnt-receptor complex, the signalosome, can be promoted by endocytosis, while at the same time, endocytosis can also attenuate Wnt signalling by regulating the membrane availability of Wnt receptors (*Albrecht et al., 2021*; *Colozza and Koo, 2021*; *Wu et al., 2021*).

AP2-associated kinase 1 (AAK1) belongs to the Ark/Prk or Numb-associated (NAK) family of serine/threonine protein kinases (*Smythe and Ayscough, 2003*; *Sorrell et al., 2016*). Members of this family play an important role in regulating endocytosis in yeast, *Drosophila,* or mammals (*Smythe and Ayscough, 2003*; *Peng et al., 2009*). Human AAK1 has been shown to phosphorylate AP2M1, the μ2 subunit of the AP2 adaptor complex (*Conner and Schmid, 2002*). Phosphorylation of μ2 at Thr156 by AAK1 enhances the binding affinity of AP2 to the sorting signals of endocytosed proteins (*Ricotta et al., 2002*). AAK1 has also been shown to phosphorylate the endocytic adaptor and negative regulator of Notch signalling Numb and affect its subcellular localization (*Sorensen and Conner, 2008*). AAK1 can also interact with a membrane-tethered activated form of the Notch receptor and promote its internalization (*Gupta-Rossi et al., 2011*).

AAK1 knock-down resulted in elevated Wnt signalling in a mouse embryonic stem cell-based kinase and phosphatase siRNA screen (*Groenendyk and Michalak, 2011*). A recent study in human cells confirmed this observation and offered a mechanism. By phosphorylating AP2, AAK1 promotes endocytosis of LRP6 and thus shuts down the signalling (*Agajanian et al., 2019*). In *C. elegans*, the AAK1 orthologue SEL-5 has been genetically implicated in Notch signalling as a mutation in *sel-5* can suppress the constitutively active *lin-12*/Notch mutants. However, suppression is observed only in the *lin-12* allele activating the membrane-anchored Notch and not when the pathway is activated by expression of the intracellular domain (*Fares and Greenwald, 1999*). This hints that SEL-5 could regulate endocytosis in a manner similar to that of its mammalian counterpart. So far, no link has been made between SEL-5 and *C. elegans* Wnt signalling regulation.

In this study, we set out to analyse the potential role of the SEL-5 kinase in Wnt signalling regulation and its contribution to DPY-23/AP2M1 phosphorylation in *C. elegans*. We show that SEL-5 acts in collaboration with the retromer complex as a positive regulator of EGL-20/Wnt signalling during the migration of QL neuroblast descendants. We further demonstrate that although SEL-5 participates in DPY-23 phosphorylation, the observed role of SEL-5 in neuroblast migration is independent of its kinase activity, and the phosphorylation of DPY-23 is dispensable for Wnt-dependent migration of QL neuroblast descendants. SEL-5, together with the retromer, is further required during the active phase of excretory cell canal extension, again independently of its kinase activity. Finally, we show that the Wnt proteins CWN-1 and CWN-2, together with CFZ-2/Frizzled, promote excretory canal outgrowth while LIN-44/Wnt and LIN-17/Frizzled together define the stopping point for canal extension.

## Results
### Loss of SEL-5 affects migration of QL neuroblast descendants

Q cell descendants (Q.d) migration is dependent on EGL-20/Wnt signalling. While on the right side of the L1 larva QR divides and QR.d migrate in the default anterior direction, on the left side QL.d respond to the EGL-20 signal produced in several cells around the rectum by expressing the homeotic gene *mab-5* and migrate to the posterior. In the absence of active EGL-20 signalling, *mab-5* is not expressed and QL.d migrate to the anterior (*Salser and Kenyon, 1992*; *Figure 1A*). To visualize Q.d migration, we utilized *Pmec-7::gfp* expressing transgene *muIs32* which is, apart from other cells,

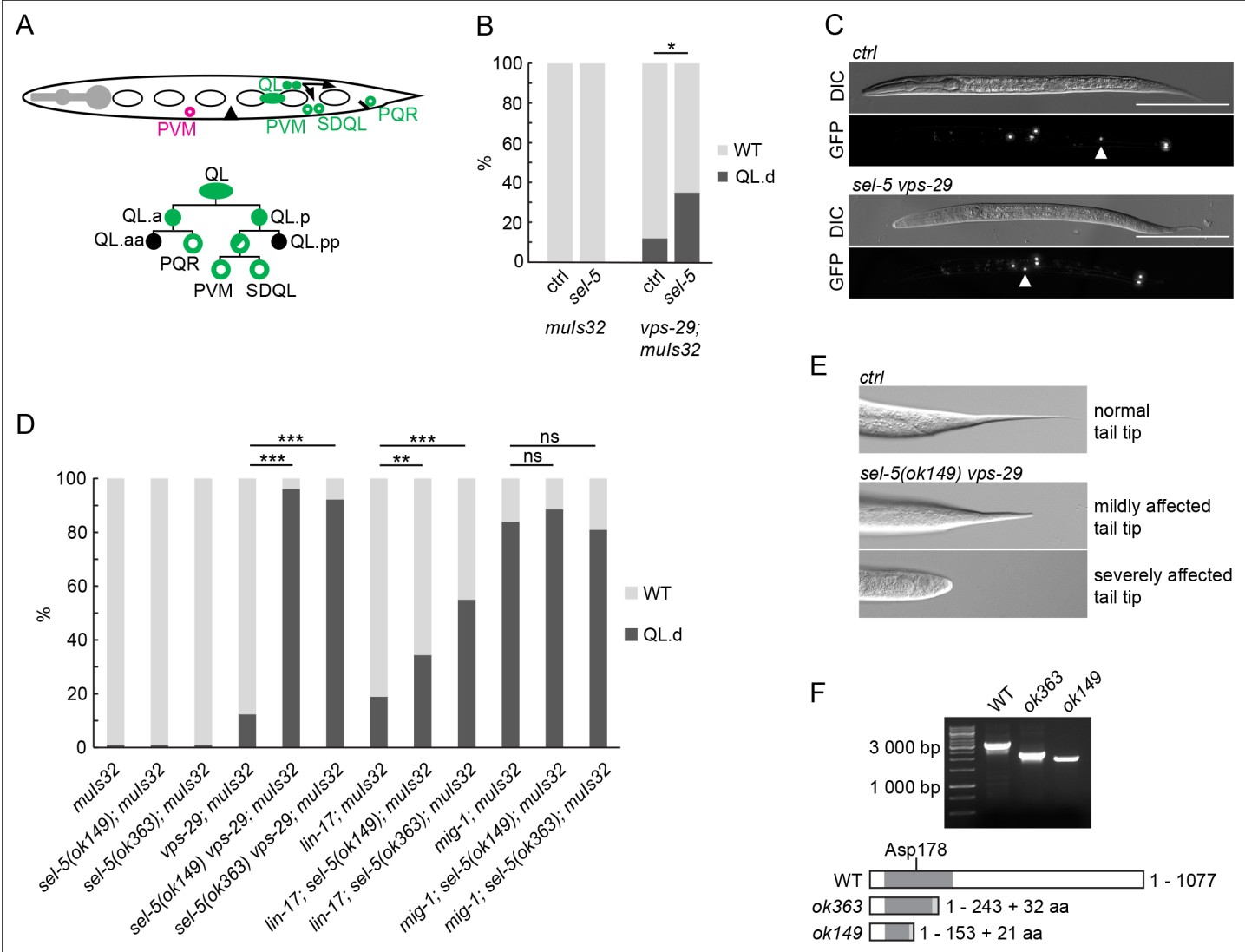

**Figure 1.** Loss of *sel-5* potentiates QL migration defect in retromer and Wnt pathway mutants. (**A**) QL neuroblast lineage and a cartoon indicating the position of terminally differentiated neurons (depicted with empty circles). (**A** has been adapted from Figure 1 of *Rella et al., 2016*.) The aberrant position of PVM neuron, as observed in QL.d migration defect, is highlighted in magenta. (**B**) RNAi against *sel-5* increases the penetrance of *vps-29* QL.d migration defect compared to the control (L4440) RNAi. No defect is observed in wild type background. (**C**) PVM position in L2 larvae of control (transgene only) and *sel-5 vps-29* double mutant animals. PVM position indicated with white arrowhead, neurons visualized by expression of *Pmec-7::gfp* transgene *muIs32*, scale bar represents 100 μm. (**D**) Mutation in *sel-5* results in increased penetrance of QL.d migration defect of *vps-29* and *lin-17*, but not *mig-1* mutants. (**E**) Examples of mild and severe alteration of tail tip morphology in *sel-5 vps-29* double mutants. (**F**) Shortened transcripts are produced from the *sel-5* locus in both *ok363* and *ok149* alleles. Potential protein products resulting from these transcripts are depicted, showing the impact of the truncation on the kinase domain (dark grey box). Extra amino acids resulting from a frameshift and thus not present in the wild type protein are also depicted (light grey box). The position of the active site is indicated. For (**B**) and (**D**), results are shown as % of WT and QL.d animals. Fisher's exact test was performed to assess the difference between the samples. Bonferroni correction for multiple testing was applied. For (**B**), a representative RNAi experiment is shown, n > 55 per condition, *p-value<0.02, additional experiments provided in *Figure 1—figure supplement 1*. For (**D**), data combined from three independent experiments are shown, n > 150 animals in total for each strain, **p-value<0.002, ***p-value<0.0002, ns, not significant.

The online version of this article includes the following source data and figure supplement(s) for figure 1:

**Source data 1.** Source gel for *Figure 1*.

**Figure supplement 1.** Additional examples of RNAi against *sel-5* in wild type and *vps-29* background.

**Figure supplement 2.** Schematic structure of the *sel-5* genomic locus.

**Table 1.** Phenotypes of *sel-5* and *vps-29* single and double mutants.

| Genotype | ALM (%)* | PLM (%)† | CAN (%) ‡ | Dye filling (%) § | Fecundity (n) ¶ |
|---|---|---|---|---|---|
| Wild type | 0.0 | 0.0 | 0.6 | 0.5 | 280 ± 41 |
| *sel-5(ok149)* | 0.0 | 0.0 | 8.9 | 4.7 | 236 ± 24 |
| *sel-5(ok363)* | 0.0 | 0,0 | 10.9 | 1.0 | 249 ± 37 |
| *vps-29(tm1320)* | 0.0 | 0.0 | 2.9 | 4.3 | 221 ± 28 |
| *sel-5(ok149) vps-29* | 29.9 | 9.1 | 17.7 | 17.4 | 132 ± 40 |
| *sel-5(ok363) vps-29* | 7.2 | 2.8 | 16.3 | 7.1 | 159 ± 23 |

*% of animals with axons of one or both ALM reversed/bipolar, n > 60.
†% of animals with axons of one or both PLM neurons reversed, n > 60.
‡% of CAN neurons located anteriorly of V3 seam cell, n > 40.
§% of animals with one or both phasmid sensilla not dyed with DiI, n > 40.
¶Average number of progeny from five hermaphrodites,± s.d.

active in AVM and PVM touch neurons, which are QR and QL descendants, respectively (***Ch'ng et al., 2003***). We performed RNA interference (RNAi) against *sel-5* in wild type background and observed no effect on QL.d migration (***Figure 1B***). To increase the sensitivity of the RNAi approach, we repeated the RNAi experiment using *vps-29(tm1320)* mutant strain. This strain harbours a null mutation in the VPS-29 retromer subunit, displays only weakly penetrant QL.d migration defect (***Coudreuse et al., 2006***), and has previously been used as a sensitized background to uncover genes required for Q.d migration (***Harterink et al., 2011b***; ***Lorenowicz et al., 2014***). RNAi against *sel-5* significantly increased the QL.d migration defect of *vps-29(tm1320)* (***Figure 1B***, ***Figure 1—figure supplement 1***). To verify the results obtained with RNAi, we assessed the QL.d migration in *sel-5(ok363)* and *sel-5(ok149)* mutants. Similarly to RNAi results, mutation of *sel-5* alone did not affect QL.d migration while *sel-5 vps-29* double mutants displayed an almost fully penetrant defect (***Figure 1C and D***).

The loss of *vps-29* affects EGL-20/Wnt signalling at the level of Wnt production (***Yang et al., 2008***). We further tested whether loss of *sel-5* could enhance partially penetrant defects in QL.d migration in mutants acting at a different step of the EGL-20/Wnt pathway. We tested two Frizzled mutants which both serve to receive the EGL-20 signal in the QL, albeit to a different extent (***Zinovyeva et al., 2008***). The loss of *sel-5* strongly increased the weak QL.d migration defect of the *lin-17*/Frizzled mutants but did not affect the already more penetrant QL.d defect of *mig-1*/Frizzled mutant animals (***Figure 1D***).

We next asked whether loss of *sel-5* in the *vps-29* background affects other Wnt-dependent processes such as the polarization of the ALM and PLM neurons, CAN neuron migration, or T cell polarity. Polarization of ALM and PLM is regulated by several Wnt proteins (***Pan et al., 2006***; ***Prasad and Clark, 2006***; ***Hilliard and Bargmann, 2006***), CAN neuron migration is governed predominantly by CWN-2 with a minor contribution from CWN-1 and EGL-20 (***Zinovyeva and Forrester, 2005***; ***Zinovyeva et al., 2008***) and T cell polarity depends on the LIN-44/Wnt signal (***Herman et al., 1995***). We observed weakly penetrant ALM polarization defects in both *sel-5(ok363) vps-29* and *sel-5(ok149) vps-29* double mutants, a very mild PLM polarization defect was detected in a strain carrying the *ok149* allele (***Table 1***). In addition, mildly penetrant anterior displacement of the CAN neuron was observed in the double mutants, while approximately 17% of the double mutant animals carrying the *ok149* allele did not correctly form the phasmid sensillum, which indicates defects in T cell polarization (***Table 1***). Apart from these Wnt-dependent phenotypes, we also observed reduced brood size in the double mutants compared to controls (***Table 1***), defective tail formation (***Figure 1E***), and a progressive egg-laying defect leading to a bag-of-worms phenotype.

The penetrance of the observed phenotypes was not the same in the two *sel-5* alleles. This was unexpected as both alleles carry deletions that hit the kinase domain of SEL-5 protein. We were able to amplify shorter than wild type transcripts from both *sel-5(ok363)* and *sel-5(ok149)* animals using primers in exon 1 and exon 17 (***Figure 1F***, ***Figure 1—figure supplement 2***). Sequencing of the *ok363* transcript revealed that amino acids 1–243 of the wild type protein followed by additional 32 amino acids would be translated due to a frameshift and a premature stop codon. This protein encompasses a larger part of the SEL-5 kinase domain including the active site. In agreement with data published

by *Fares and Greenwald, 1999*, sequencing of the *ok149* transcript revealed that it can give rise to a protein containing the first 153 amino acids of SEL-5 followed by 21 additional amino acids and thus is devoid of the active site (*Figure 1F*). If such truncated proteins are present in the mutants, either one or both may possess some residual activity that is responsible for the observed phenotypic differences.

## SEL-5 is required in EGL-20/Wnt-producing cells to direct QL.d migration

The asymmetric migration of the Q neuroblast is regulated both cell-autonomously and non-autonomously. To analyse the tissue-specific requirement of SEL-5 in QL.d migration, we expressed

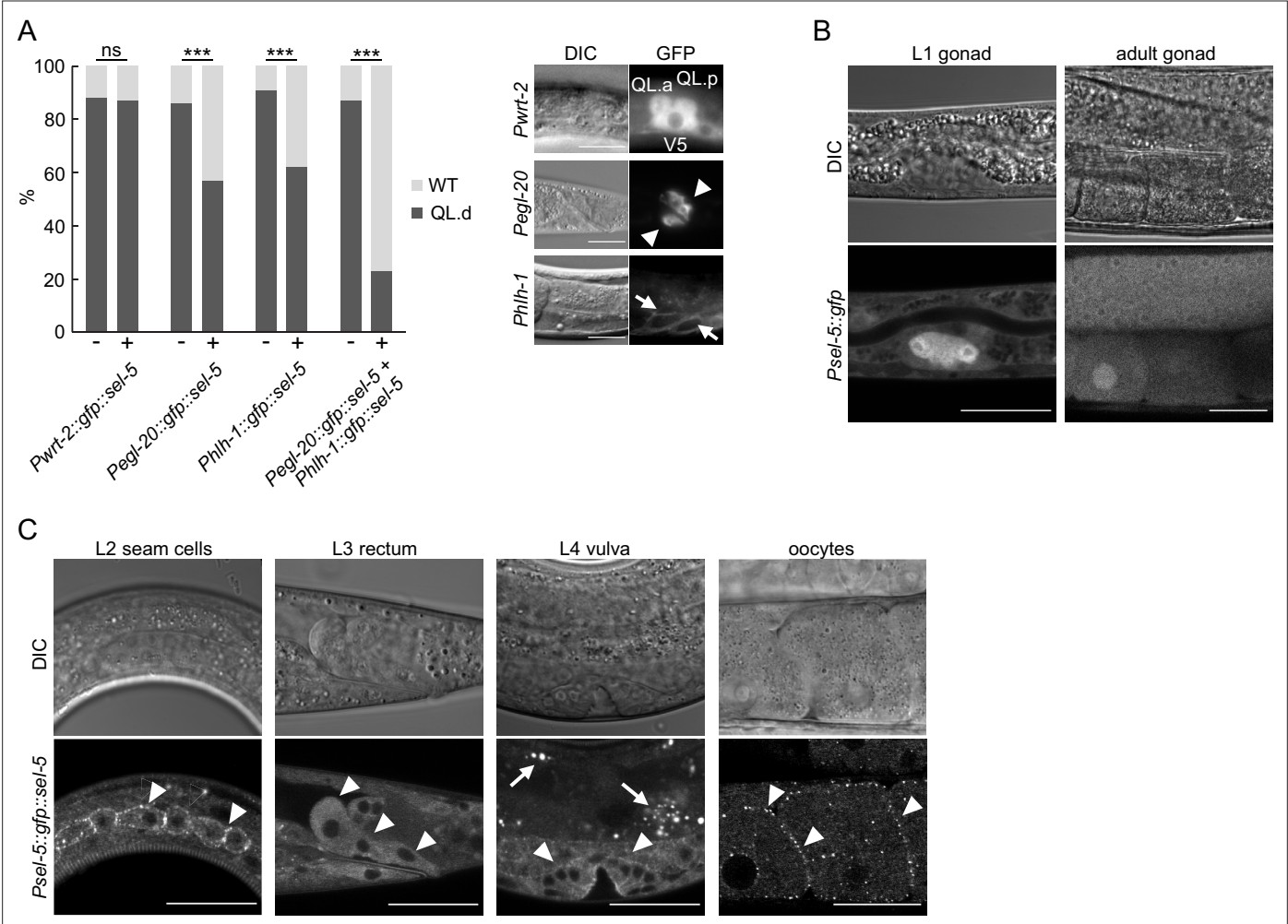

**Figure 2.** SEL-5 is expressed in multiple tissues and is required cell non-autonomously for QL.d migration. (**A**) Transgenic rescue of the QL.d migration defect. *sel-5* was expressed under the control of various promoters from an extrachromosomal array in *sel-5 vps-29; muIs32* background and the effect of such expression on QL.d migration was quantified. Comparison was made between animals carrying the transgene and their siblings which have lost the transgenic array. The expression of each transgene in the expected tissue is shown on the right. Results are shown as % of WT and QL.d animals, data are combined from three independent experiments, n > 100 animals in total for each condition. Fisher's exact test was performed to assess the difference between the samples. Bonferroni correction for multiple testing was applied, ***p-value<0.0001, ns, not significant. Results from additional independent transgenic strains are provided in *Figure 2—figure supplement 1*. (**B**) GFP expression driven by an endogenous *sel-5* promoter in the gonad of L1 and adult animals. (**C**) Expression of GFP::SEL-5 fusion protein driven by endogenous *sel-5* promoter in various larval and adult tissues. Localized GFP::SEL-5 expression is indicated by white arrowheads, white arrows point to autofluorescent signal from gut granules. The scale bar represents 20 μm in (**A–C**).

The online version of this article includes the following figure supplement(s) for figure 2:

**Figure supplement 1.** Transgenic rescue of the QL.d migration defect.

full-length SEL-5 tagged with GFP at its N-terminus under the control of different tissue-specific promoters in a *sel-5 vps-29* background and assayed QL.d migration. The expression of SEL-5 from the *egl-20* promoter (*Coudreuse et al., 2006*) resulted in a significant rescue of the QL.d migration defect of *sel-5 vps-29* double mutants albeit not to the level of *vps-29* single mutants (*Figure 2A*, *Figure 2—figure supplement 1*). No rescue was obtained when SEL-5 was expressed from the *wrt-2* promoter that is active in the seam cells and the Q cells (*Aspöck et al., 1999*; *Middelkoop et al., 2012*). These data suggest that *sel-5* is required in the EGL-20/Wnt-producing cells to control QL.d migration. However, a similar level of rescue as with *egl-20* promoter was obtained also when SEL-5 was expressed from *hlh-1* promoter specific for the body wall muscle cells (*Krause et al., 1990*; *Harfe et al., 1998*). When SEL-5 was simultaneously expressed from the *egl-20* and *hlh-1* promoters, the rescue almost reached the background frequency of the QL.d defect, caused by *vps-29* alone (*Figures 1D and 2A*, *Figure 2—figure supplement 1*). Apart from the rectal epithelial cells, *egl-20* expression was also detected in posterior ventral body wall muscle quadrants VL23 and VR24 (*Harterink et al., 2011a*); therefore, it is possible that *hlh-1* promoter-driven expression of SEL-5 in these muscle quadrants is responsible for the observed rescue, and only the combined expression from all EGL-20-producing cells is sufficient to drive a complete rescue. Alternatively, the observed rescue pattern could indicate that SEL-5 is required in both the Wnt-producing and in the muscle cells.

To confirm that the tissues identified above in the rescue experiments are tissues with a genuine *sel-5* expression, we set to determine the endogenous *sel-5* expression pattern that has not been previously analysed. The only published expression data (*Fares and Greenwald, 1999*) were based on the expression from *sel-12* instead of *sel-5* promoter. We generated GFP knock-in strains by inserting GFP into the *sel-5* genomic sequence using the SEC cassette approach (*Dickinson et al., 2015*). We obtained two strains, one expressing a transcriptional P*sel-5::gfp* reporter (*Figure 2B*) and one, after excision of the SEC cassette, an N-terminally tagged GFP::SEL-5 under the control of its own promoter (P*sel-5::gfp::sel-5*) (*Figure 2C*). Microscopic analyses of both strains revealed that *sel-5* is expressed broadly, but at a low level judged by the intensity of the GFP signal. The most prominent expression was observed in the gonad from the beginning of its development (*Figure 2B*). Expression was further observed in the developing oocytes, in the vulva, in epidermal cells, and most notably, in the rectal epithelial cells that are known to produce EGL-20/Wnt (*Whangbo and Kenyon, 1999*; *Figure 2B and C*). We could not detect *sel-5* expression in muscles, although we cannot exclude that a low level of expression, below the detection limit using the endogenous locus-tagging approach, is present. Interestingly, oocytes and epidermal seam cells expressing the GFP::SEL-5 fusion protein revealed its distinct subcellular localization. Protein localized to punctate structures located close to the cell surface (*Figure 2C*). This pattern resembles the subcellular distribution of human AAK1 (*Conner and Schmid, 2002*), hinting that SEL-5 could be involved in regulating intracellular transport similar to AAK1.

## SEL-5 does not affect MIG-14 endocytosis

The SEL-5 orthologue AAK1 has been implicated in endocytosis regulation (*Conner and Schmid, 2002*; *Ricotta et al., 2002*). The obvious candidate that could be affected by SEL-5-dependent endocytosis in Wnt-producing cells is the Wnt cargo receptor MIG-14/Wls. Human AAK1 can phosphorylate the AP2 subunit μ2 (AP2M1) (*Conner and Schmid, 2002*; *Ricotta et al., 2002*) and thus increase the affinity of AP2 to cargo molecules (*Ricotta et al., 2002*). It is conceivable that SEL-5 could regulate MIG-14/Wls trafficking at the level of AP2-dependent endocytosis. The *C. elegans* μ2 subunit of the AP2 complex, DPY-23 (also known as APM-2), has been shown to participate in the internalization of MIG-14 (*Pan et al., 2008*; *Yang et al., 2008*). We first tested whether loss of *sel-5* expression has any effect on the level of phosphorylation of DPY-23. A phospho-specific antibody recognizing phosphorylated threonine T160 (T156 in mammalian AP2M1) of DPY-23 (*Hollopeter et al., 2014*) revealed a decrease in the level of phosphorylated endogenous DPY-23 in *sel-5* mutants compared to wild type animals (*Figure 3A*, *Figure 3—source data 1*). Similarly, T160 phosphorylation was reduced on overexpressed GFP-tagged DPY-23 (*Figure 3B*, *Figure 3—source data 1*). Next, we assessed MIG-14 levels and localization in *sel-5 vps-29* mutants using a transgene expressing a functional MIG-14::GFP protein (*Lorenowicz et al., 2014*). However, the levels of MIG-14::GFP did not change significantly in *sel-5 vps-29* compared to the *vps-29* single mutant (*Figure 3C and D*, *Figure 3—source data 1*). Levels of MIG-14::GFP in *sel-5* single mutants were variable but on average comparable to the wild

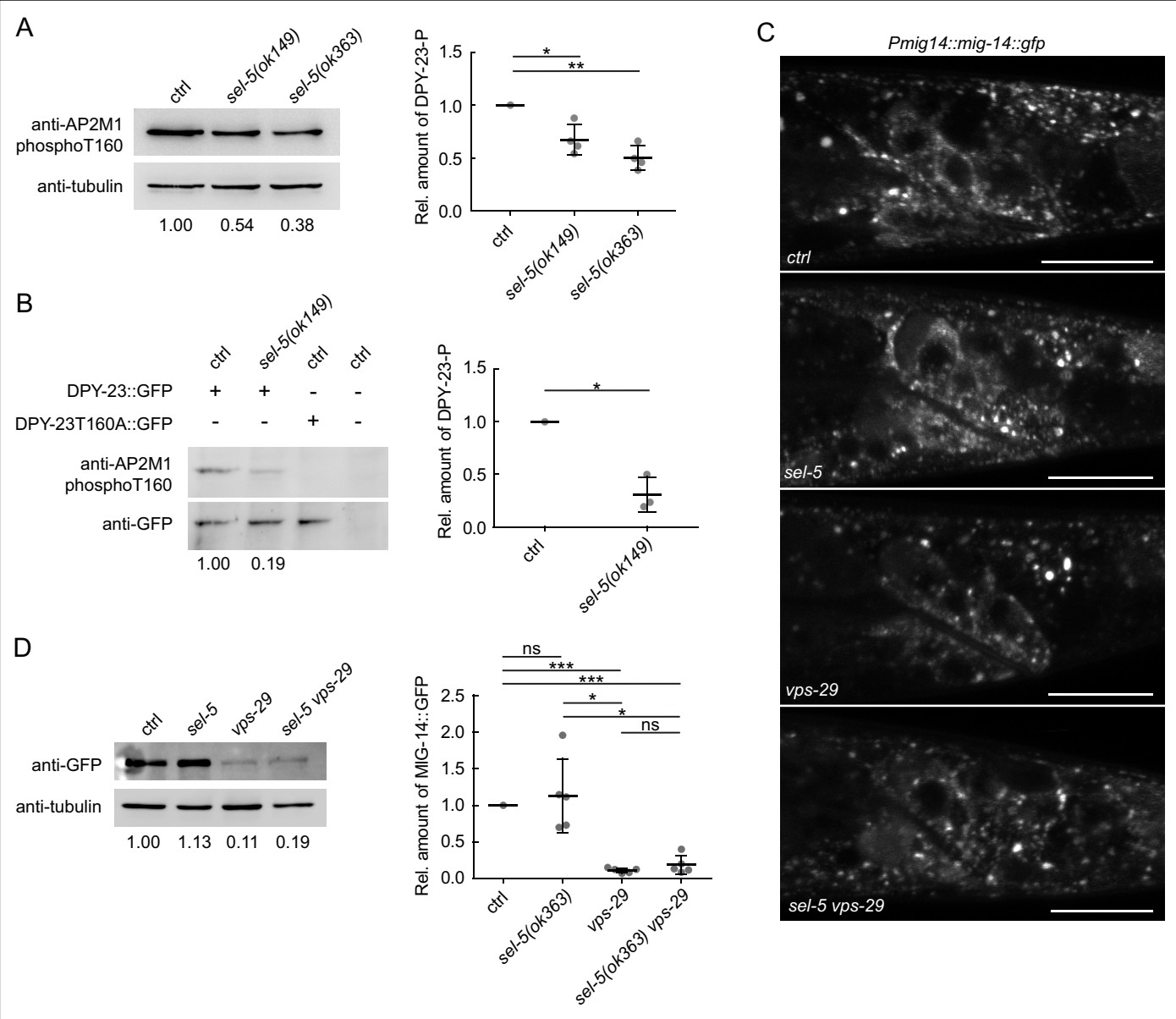

**Figure 3.** SEL-5 alters the phosphorylation status of DPY-23 but does not affect MIG-14 levels. (**A**) The level of DPY-23 phosphorylation at position T160 is reduced in *sel-5* mutant animals. Phosphorylation was detected by western blot analysis in lysates from a population of L4/young adults of indicated strains using phospho-specific antibody against human AP2M1. Band intensities were normalized to an alpha tubulin loading control and compared to the control sample (sample/control ratio indicated below each lane). A representative Western blot example is shown on the left, relative quantification of four independent experiments is shown on the right. (**B**) Level of DPY-23::GFP phosphorylation is reduced in *sel-5* mutants. Phosphorylation at T160 of GFP-tagged DPY-23 was detected by western blot analysis in lysates from a population of L4/young adults using phospho-specific antibody against human AP2M1. Strains expressing either no GFP fusion protein or a GFP-tagged DPY-23T160A mutant protein were included as controls. Band intensities were normalized to a GFP signal intensity and compared to the control sample (DPY-23::GFP in wild type background), sample/control ratio is indicated below each lane. A representative western blot example is shown on the left, relative quantification of three independent experiments is shown on the right. (**C**) L2/L3 animals expressing MIG-14::GFP from the *huSi2* transgene were imaged using a confocal microscope. The posterior part of the body with Wnt-expressing cells is shown. Images are maximum projections of four consecutive z-sections taken at 0.3 μm interval. Anterior to the left, dorsal up, scale bar represents 20 μm. (**D**) Western blot analysis of MIG-14::GFP levels expressed from a *huSi2* transgene in various mutant backgrounds. MIG-14::GFP was detected in lysates from synchronized populations of L1 larvae of the indicated strains. Band intensities were normalized to an alpha tubulin loading control and compared to the control sample (sample/control ratio indicated below each lane). A representative western blot example is shown on the left, relative quantification of five independent experiments is shown on the right. In (**A**, **B**, **D**), error bars represent mean ± s.d., statistical significance was assessed by unpaired two-tailed Student's *t*-test for samples with unequal variance, *p-value<0.05, **p-value<0.01, ***p-value<0.001, ns, not significant. Source data for (**A**, **B**, **D**) provided in *Figure 3—source data 1*.

*Figure 3 continued on next page*

*Figure 3 continued*

The online version of this article includes the following source data and figure supplement(s) for figure 3:

**Source data 1.** Source blots for *Figure 3*.

**Figure supplement 1.** MIG-14 does not re-localize to the plasma membrane in *sel-5* or *sel-5 vps-29* mutants.

type. We therefore concluded that the significant increase in QL.d migration defects in *sel-5 vps-29* compared to *vps-29* cannot be attributed to changes in MIG-14/Wls levels. Importantly, MIG-14::GFP did not re-localize to the plasma membrane in *sel-5* or *sel-5 vps-29* mutants, in striking contrast to the MIG-14::GFP behaviour in animals treated with *dpy-23* RNAi (*Figure 3—figure supplement 1*). This excludes the possibility that the Wnt-related phenotypes in *sel-5 vps-29* mutants arise from a defect in MIG-14 internalization.

## SEL-5 helps shape the EGL-20 gradient

Our tissue-specific rescue experiments revealed that for a full rescue, simultaneous *sel-5* expression is necessary from both *egl-20* and *hlh-1* promoters. SEL-5 could therefore regulate endocytosis along the EGL-20 transport route, thus shaping the EGL-20 gradient. To test this hypothesis, we endogenously tagged EGL-20 with GFP and assessed Wnt gradient formation in control and mutant backgrounds. This approach has been previously used and revealed that extracellular EGL-20 can be detected in the form of distinct puncta that most likely represent EGL-20 bound to Frizzled (*Pani and Goldstein, 2018*). We detected EGL-20 puncta in all strains tested, albeit with different frequencies (*Figure 4A–C*). While the number of EGL-20 puncta in *sel-5* mutants was not significantly different from that in control animals, in *sel-5 vps-29* mutants the number of puncta was substantially reduced. However, statistical evaluation did not reveal significant differences between *vps-29*, *sel-5 vps-29*, and *vps-35*; although a decreasing trend in puncta number was visible (*Figure 4C*). In *vps-35*, the EGL-20 gradient was shown to be highly reduced or absent (*Coudreuse et al., 2006*; *Harterink et al., 2011b*) and was used for comparison. We also compared the signal intensity in the EGL-20 spots among the different strains but no significant differences were detected (*Figure 4D*). Finally, we assessed the length of the gradient by measuring how far from the rectum the EGL-20 puncta can be detected. While EGL-20 gradient formation in neither *sel-5* nor *vps-29* was significantly affected, the EGL-20 gradient was shorter in *vps-29 sel-5* double mutants than in wild type controls (*Figure 4E*). Our data thus support the notion that *sel-5* plays a subtle role in EGL-20 gradient formation.

## SEL-5-associated phenotypes are independent of DPY-23 phosphorylation

Although endocytic cargo other than MIG-14 may be affected in *sel-5 vps-29* mutants, we could not exclude the possibility that the observed phenotypes are a consequence of defects in other mechanisms unrelated to DPY-23 phosphorylation. To test this hypothesis, we repeated the rescue experiment presented in *Figure 2A* now with SEL-5 carrying either K75A or D178A point mutations. Position D178 corresponds to D176 of human AAK1 and is part of the conserved HRD motif in the catalytic loop, K75 corresponds to K74 in AAK1 and is predicted to affect ATP binding (*Figure 5—figure supplement 1*). In AAK1, both mutations abolished its ability to phosphorylate AP2M1 (*Conner and Schmid, 2003*). When expressed under the control of the *egl-20* promoter, the mutant SEL-5 proteins could still rescue the QL migration phenotype (*Figure 5A*, *Figure 5—figure supplement 2*). This indicates that SEL-5 kinase activity is not responsible for the phenotypes observed in *sel-5 vps-29* mutants. To test this further, we next asked whether phosphorylation at the T160 position of DPY-23 is necessary for QL.d migration. To this end, we utilized the *dpy-23(mew25)* allele harbouring T160A point mutation (G. Hollopeter and G. Beacham, unpublished). Characterization of *dpy-23(mew25)* animals revealed that homozygous mutants are viable; they are not dumpy and look superficially wild type. This indicates that the mutants do not suffer from a gross endocytosis defect, even though no DPY-23 phosphorylation was detected in *dpy-23(mew25)* animals (*Figure 5B*, *Figure 5—source data 1*). In compliance with previous findings using *dpy-23* alleles that harbour mutations changing the T160 amino acid (*Hollopeter et al., 2014*; *Partlow et al., 2019*), *mew25* is able to rescue the 'jowls' phenotype of *fcho-1(ox477)* mutants (*Figure 5C*). FCHO-1 is a member of the muniscin family of proteins and a proposed allosteric activator of

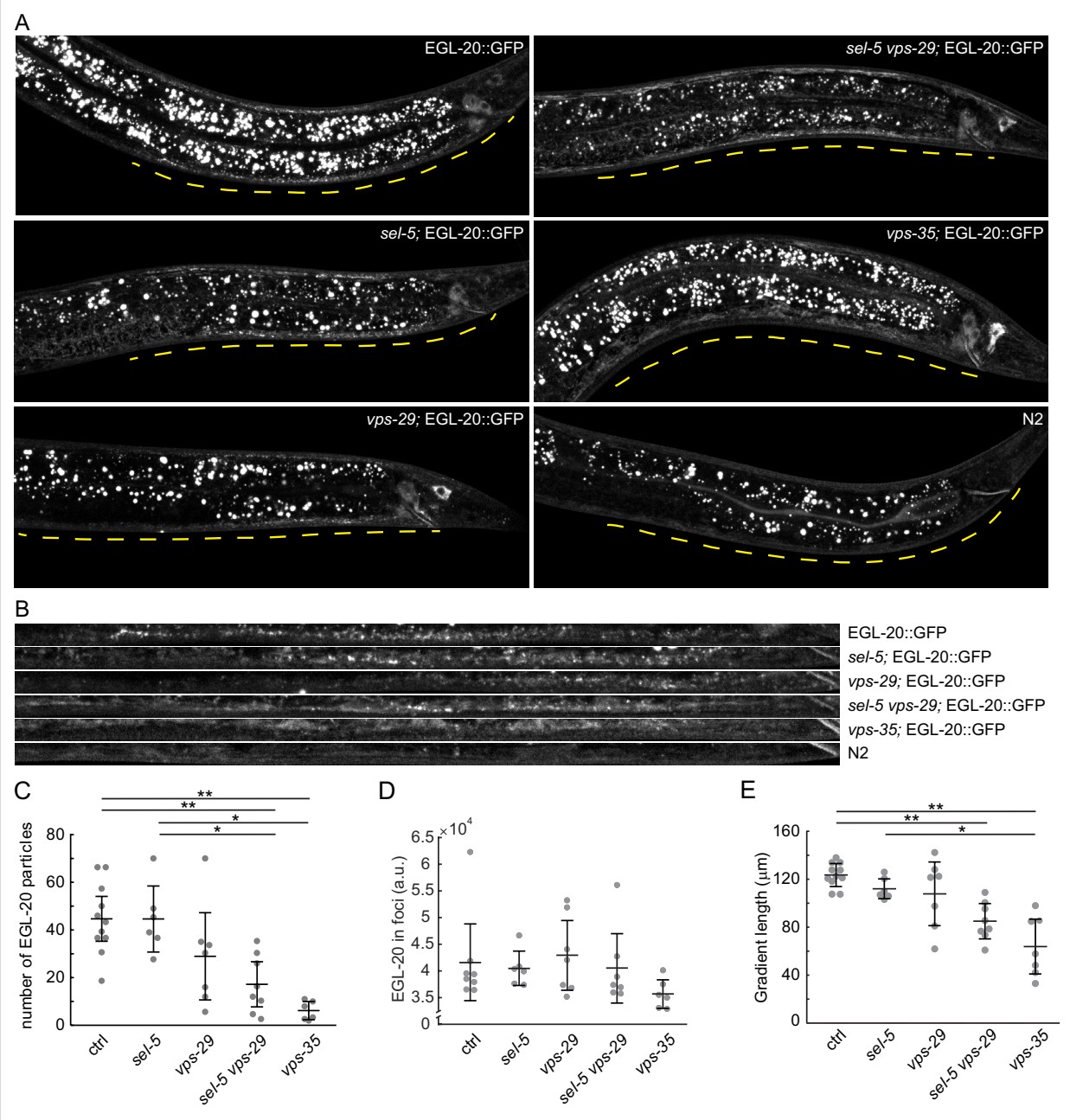

**Figure 4.** EGL-20 gradient formation. (**A**) Endogenously tagged EGL-20::GFP was visualized in various backgrounds. Images were acquired with spinning disc microscopy using L2/L3 animals and represent maximum projections of seven consecutive z-sections captured at 1 μm interval. A distance of 150 μm from the rectum of each animal is indicated by a yellow line. *sel-5(ok149)* allele was used in *sel-5* single and double mutants. (**B**) A curved line of 150 μm length (measured from the rectum) and 3.9 μm width was fitted with the ventral side of each animal presented in (**A**) and the selected region was straightened using the 'Straighten' function in Fiji for easier comparison. (**C**) Number of EGL-20 particles in various mutant backgrounds. Identical selection as in (**B**) was applied to all samples and the number of clearly visible puncta was manually counted in blinded images. Counting was repeated three times for each sample and the number of puncta was averaged and plotted as a single data point. (**D**) Mean signal intensity in EGL-20 particles was measured in the same selections as in (**C**). (**E**) The length of the EGL-20 gradient was assessed in the same images as in (**C**). The distance from the rectum to the most distant clearly recognizable EGL-20 particle was measured in blinded images. Measurement was repeated three times; the values were averaged and plotted as a single data point. For (**C, D, E**), Wilcoxon rank sum test was performed to assess statistical significance. Bonferroni correction for multiple testing was applied. *p-value<0.005, **p-value<0.001, error bars represent 95% confidence interval, only statistically significant differences shown in (**C**) and (**E**), and no statistically significant differences revealed in (**D**).

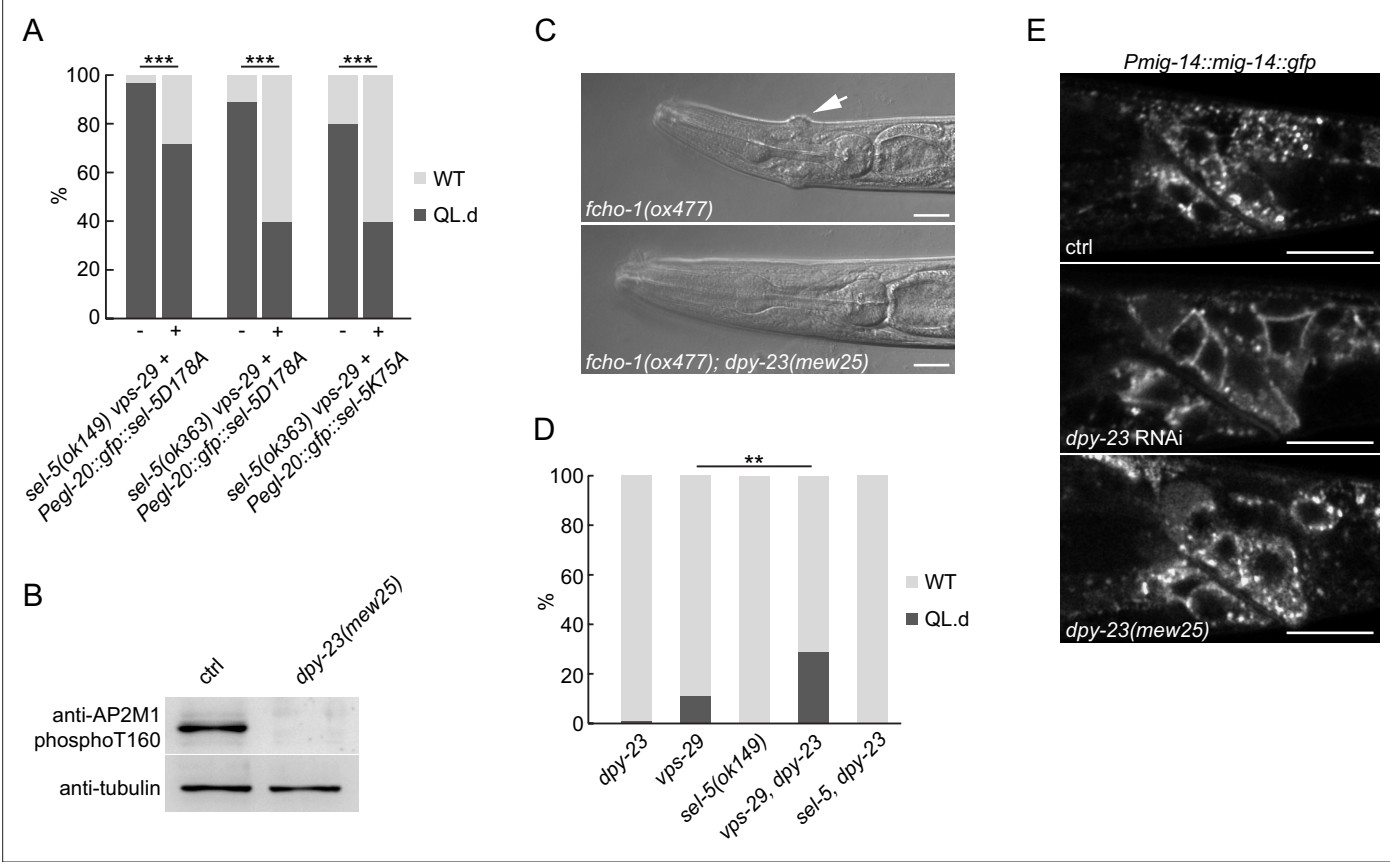

**Figure 5.** SEL-5 kinase activity and DPY-23 phosphorylation are not required for QL.d migration. (**A**) Transgenic rescue of the QL.d migration defect with kinase-inactive SEL-5. D178A or K75A SEL-5 mutant protein was expressed from *egl-20* promoter in *sel-5 vps-29; muIs32* mutant background from an extrachromosomal array and the QL.d migration defect was quantified in animals carrying the transgene and their siblings which have lost the transgenic array. Results are shown as % of WT and QL.d animals, data are combined from three independent experiments, n > 100 animals in total for each condition. Fisher's exact test was performed to assess the difference between the samples. Bonferroni correction for multiple testing was applied, ***p-value<0.0001. Results from additional independent transgenic strains are provided in *Figure 5—figure supplement 2*. (**B**) DPY-23 phosphorylation at position T160 is absent in *dpy-23(mew25)* mutant animals. Phosphorylation was detected by western blot analysis in lysates from a population of L4/ young adults of indicated strains using phospho-specific antibody against human AP2M1. Band intensities were normalized to an alpha tubulin loading control. (**C**) *dpy-23(mew25)* rescues the 'jowls' phenotype (white arrow) of *fcho-1(ox477)* animals. Heads of adult animals are shown, anterior to the left, the scale bar represents 10 µm. (**D**) The absence of DPY-23 T160 phosphorylation does not result in QL.d migration defect. The presence of *dpy-23(mew25)* allele carrying T160A substitution either alone or in combination with *sel-5* does not significantly contribute to the QL.d migration defect. Results are shown as % of WT and QL.d animals, data combined from at least three independent experiments are shown, n > 180 animals in total for each strain. Fisher's exact test was performed to assess the difference between the samples, **p-value<0.01. (**E**) MIG-14::GFP does not relocalize to the plasma membrane in *dpy-23(mew25)* animals. L3 animals expressing MIG-14::GFP from the *huSi2* transgene were imaged using a confocal microscope. The posterior part of the body with Wnt-expressing cells is shown. Images are maximum projections of five consecutive z-sections taken at 0.3 µm interval. Anterior to the left, dorsal up, the scale bar represents 20 µm. Source data for (**B**) provided in *Figure 5—source data 1*.

The online version of this article includes the following source data and figure supplement(s) for figure 5:

**Source data 1.** Source blots for *Figure 5*.

**Figure supplement 1.** Sequence alignment of the kinase domain of SEL-5 and human AAK1.

**Figure supplement 2.** Transgenic rescue of QL.d migration defect by kinase-inactive SEL-5.

AP2 (*Hollopeter et al., 2014*). We then asked whether complete loss of DPY-23 phosphorylation would lead to a QL.d migration defect. Animals carrying *dpy-23(mew25)* alone or in combination with *sel-5(ok149)* did not show any defects in QL.d migration (*Figure 5D*). A variable increase in QL.d migration defects was observed in *vps-29;dpy-23(mew25)* animals, but the penetrance of the defect never reached levels comparable to those in *sel-5 vps-29* mutants (*Figures 1D and 5D*). Furthermore, MIG-14::GFP did not re-localize to the plasma membrane in *dpy-23(mew25)* similar to the *sel-5* mutants and unlike in *dpy-23* RNAi animals (*Figure 5E*). Together, these observations

strongly suggest that the role of SEL-5 in the regulation of QL.d migration is not dependent on its kinase activity and, moreover, that DPY-23 phosphorylation at T160 is not a major regulatory event in this process.

## The outgrowth of excretory cell canals is impaired in *sel-5 vps-29* mutants

Apart from the QL.d migration defect in the *sel-5 vps-29* double mutants, in some animals, we unexpectedly noticed a severe shortening of the posterior canals of the excretory cell that prompted us to analyse this phenotype in more detail. The excretory cell (also called the excretory canal cell) is a large H-shaped cell required for osmoregulation (*Buechner et al., 1999*; *Liégeois et al., 2007*). The excretory canal cell body is located near the posterior bulb of the pharynx and four excretory canals emanate from the cell body, two short ones directed to the anterior and two posterior canals extending to the rectum (*Figure 6A*). To assess the morphology of the excretory cell, we expressed GFP under the control of *pgp-12* promoter which is active exclusively in the excretory cell (*Zhao et al., 2004*) and analysed the length of the canals in late L4 or early adult animals. While the posterior canal length in *sel-5* or *vps-29* single mutants was indistinguishable from the wild type controls, in more than 60% of the *sel-5 vps-29* mutants the posterior canals stopped at various positions anterior to the rectum (*Figure 6A and B*). A similar effect, albeit with lower penetrance, was observed when assessing the anterior canal length (*Figure 6C*). Interestingly, the posterior canal on the right side of the animal was significantly more affected than its counterpart on the left side in *sel-5 vps-29* double mutants carrying the *ok149* allele (*Figure 6D*). A similar trend was observed in *sel-5(ok363) vps-29* animals, although the difference observed there did not reach statistical significance (*Figure 6D*).

Active growth of the canals starts during embryogenesis and continues during L1. After that the canals passively grow with the growing animal (*Fujita et al., 2003*). Comparison of canal lengths at several time points within a 24 hr interval after hatching revealed that in *sel-5 vps-29* double mutants the posterior canals of the excretory cell are shorter already at the time of hatching and the growth defect prevails throughout larval development (*Figure 6E*).

Excretory canal shortening has not previously been reported for any of the retromer components. Therefore, we tested whether this phenotype is specific for the *vps-29* retromer subunit or whether the whole retromer complex is required together with *sel-5* for proper excretory canal extension. We performed RNAi against *vps-35*, *vps-26*, *snx-1*, and *snx-3* in *sel-5(ok149)* background and assessed posterior canal length. Except for *snx-1* RNAi, significant shortening was observed with RNAi against all other retromer subunits, with *vps-35* showing the strongest phenotype (*Figure 6F*). These data confirm that the whole retromer complex acts together with *sel-5* to control excretory canal outgrowth.

Next, we tested in which tissue SEL-5 activity is required for excretory cell outgrowth. We expressed SEL-5 from either *pgp-12* (excretory cell), *hlh-1* (body wall muscle cells), or *col-10* (hypodermis) promoters in *sel-5 vps-29* double mutants and checked for any rescue. As shown in *Figure 6G*, the expression of SEL-5 from the *pgp-12* promoter almost fully rescued the excretory cell shortening while the expression of SEL-5 from the muscle-specific *hlh-1* promoter did not affect the excretory cell phenotype of the *sel-5 vps-29* double mutants. As in the case of the QL.d migration defect, the excretory canal length was efficiently rescued also by the SEL-5D178A variant expressed from *pgp-12* promoter arguing that SEL-5 kinase activity is not required for canal extension (*Figure 6G*). Expression of VPS-29 in the excretory cell partially rescued canal shortening, suggesting that SEL-5 and the retromer complex act together within the same cell to regulate canal length. Interestingly, the expression of SEL-5 from the *col-10* promoter, which is active in the hypodermis (*Spencer et al., 2001*), also resulted in a significant rescue of the excretory canal shortening. This suggests that SEL-5 activity in both the excretory cell and the hypodermis contributes to the regulation of canal outgrowth.

To support our finding that the role of SEL-5 in excretory canal outgrowth is independent of DPY-23 phosphorylation, we analysed the length of the posterior canal in *dpy-23(mew25)* mutant animals. No canal shortening was observed in animals carrying *dpy-23(mew25)* alone (*Figure 6H*), while only very mild shortening was occasionally observed in either *sel-5; dpy-23(mew25)* or *vps-29; dpy-23(mew25)*. These observations are consistent with a role of SEL-5 in excretory canal outgrowth that is not dependent on DPY-23 phosphorylation at T160.

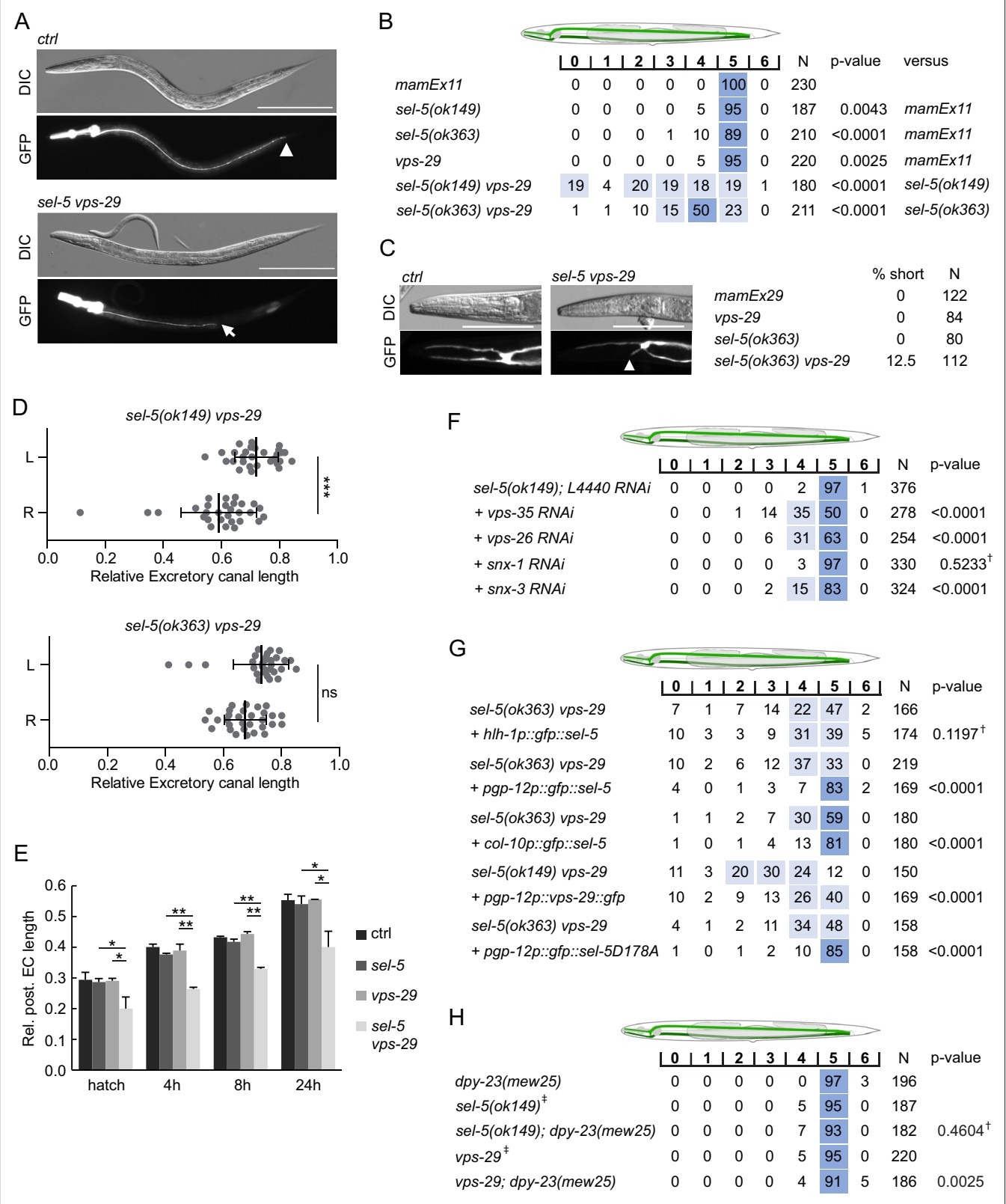

**Figure 6.** *sel-5* cooperates with the retromer complex to regulate the length of excretory cell canals. (**A**) Posterior canals of the excretory cell are significantly shortened in *sel-5 vps-29* mutants. The excretory cell was visualized by *Ppgp-12::gfp* expression from a *mamEx11* transgene. The scale bar represents 200 μm. (**B**) Quantification of posterior canal outgrowth defects. The outgrowth of the posterior canal of the excretory cell was quantified by dividing the region between the posterior bulb of the pharynx to the tip of the tail into seven segments. The percentage of canal arms terminating

*Figure 6 continued on next page*

*Figure 6 continued*

in each segment is indicated. Segment scoring 50% or higher was highlighted in dark blue, segment scoring 15–49% was highlighted in light blue for easier orientation. All strains contained the *mamEx11* transgene to visualize the excretory cell. The statistical significance of differences between the strains was analysed using Fisher's exact test for a 2 × 3 table. For the test, data from segments 0–4 were pooled into one category ('shorter') and data from segments 5 and 6 were used as the two other categories ('normal' and 'longer', respectively). The comparison was made either to the control strain or to the more severe single mutant in the case of double mutant strains. Bonferroni correction for multiple testing was applied. (**C**) Anterior canals of the excretory cell are shortened in *sel-5 vps-29* double mutants. The excretory cell was visualized by *Ppgp-12::gfp* expression from a *mamEx29* transgene. The scale bar represents 100 μm. (**D**) Posterior canals on the right side of the animals are more severely affected compared to their left counterparts. The length of the posterior canal on each side of the animal was measured and normalized to the length of the whole body of the animal. Paired Student's *t*-test was used to assess the significance of the difference between the two sides, ***p-value<0.001, ns, not significant. (**E**) Posterior excretory canals are shorter already at the time of hatching in *sel-5 vps-29* mutants. The dynamics of posterior canal outgrowth was assessed by measuring the canal length and normalizing it to the total body length at hatching and at three time points during early larval development. Results are presented as mean + s.d. of at least 30 canals for each condition. Unpaired two-tailed Student's *t*-test was performed to assess the difference between the samples, *p-value<0.05, **p-value<0.01. (**F**) Loss of either *vps-35*, *vps-26*, or *snx-3* retromer component expression induces posterior canal shortening in *sel-5* mutants. Canal outgrowth was scored as in (**B**). (**G**) Both cell-autonomous and non-autonomous expression of *sel-5* rescues excretory canal shortening in *sel-5 vps-29* mutants. SEL-5 was expressed from an extrachromosomal array under the control of *pgp-12*, *hlh-1* or *col-10* promoters. Canal outgrowth was scored as in (**B**) and a comparison was made between animals carrying the array and their siblings which have lost the transgenic array. All strains contained also the *mamEx29* extrachromosomal array to visualize the excretory canal. (**H**) T160 phosphorylation of DPY-23 is not required for posterior excretory canal outgrowth. Canal outgrowth was scored as in (**B**) in strains containing *dpy-23(mew25)* allele carrying T160A substitution. Comparison was made between the double mutants and either *sel-5* or *vps-29* single mutant. For (**F–H**) †differences not significant, ‡same data as in (**B**).

## Wnt-dependent signalling is required to establish proper excretory cell canal length

We were interested to see whether the observed shortening of the excretory canals in *sel-5 vps-29* mutants could be a consequence of crosstalk with the Wnt signalling pathway as in the case of QL.d migration. *lin-17/Frizzled* mutants have been reported to show an overgrowth of the posterior excretory cell canals past the rectum into the tip of the tail (*Hedgecock et al., 1987*) while loss of *axl-1*, one of the two *C. elegans* Axin orthologues, resulted in ectopic branching of the posterior excretory canal without affecting canal length. Ectopic branching in *axl-1* mutants could be rescued by simultaneous loss of *bar-1*/β-catenin or *pop-1*/Tcf expression (*Oosterveen et al., 2007*). Wnt signalling thus seems to play a role in excretory canal growth, but so far, the other Wnt pathway components and their mechanism of action remained unknown. Therefore, we tested whether mutations in other Wnt pathway components could affect excretory cell growth. Among the four Wnts tested (EGL-20, LIN-44, CWN-1, CWN-2), only mutants in *lin-44*/Wnt exhibited an almost fully penetrant posterior excretory canal overgrowth phenotype, as observed in *lin-17*/Frizzled mutants (*Figure 7A and B*). Mild canal shortening was observed in *cwn-2*/Wnt and *cfz-2*/Frizzled mutants, while a very weak overgrowth phenotype could be detected in *egl-20*/Wnt mutants (*Figure 7B*). No change in excretory canal length was displayed by *cwn-1*/Wnt or *mig-1*/Frizzled mutants. Interestingly, highly penetrant canal overgrowth was also observed in *mig-14*/Wls mutants, while loss of *mig-5*/Dishevelled or *dsh-1*/Dishevelled resulted in a partially penetrant overgrowth phenotype (*Figure 7B*). We next tested various combinations of Wnt and Frizzled mutants and found that simultaneous loss of *lin-44* and *cwn-1* or *lin-44* and *cwn-2* expression resulted in partial rescue of the canal overgrowth and the same effect was observed after simultaneous loss of *lin-17* and *cfz-2*. Surprisingly, the loss of *egl-20* and *mig-1* also partially rescued the canal overgrowth in *lin-17*, even though neither of them displayed canal shortening on its own. In contrast, *mig-1* suppressed canal shortening in *cfz-2*. The most prominent canal shortening was observed in *cwn-1; cfz-2* or *cwn-2; cfz-2* double mutants (*Figure 7B*). Apart from shortening, three instead of two posterior canal branches were detected in 11% of *cwn-1; cfz-2* and 15% of *cwn-2; cfz-2* double mutants (*Figure 7C*). These data suggest that at least two Wnt-dependent pathways are acting during the extension of the excretory canal, one involving *lin-44*, *lin-17*, *dsh-1* and *mig-5* that is responsible for determining the stopping point for the growing canal, and one involving *cwn-1*, *cwn-2*, and *cfz-2* that contributes to the canal growth.

The most striking phenotype is the overgrowth of the posterior excretory canal in *lin-17*, *lin-44*, and *mig-14* mutants. Interestingly, when we measured the length of the excretory canal within the 24 hr interval after hatching, no difference in canal length was detected between wild type controls and *lin-17*, *lin-44*, and *mig-14* mutants (*Figure 7D*). This suggests that the initial growth is not affected in

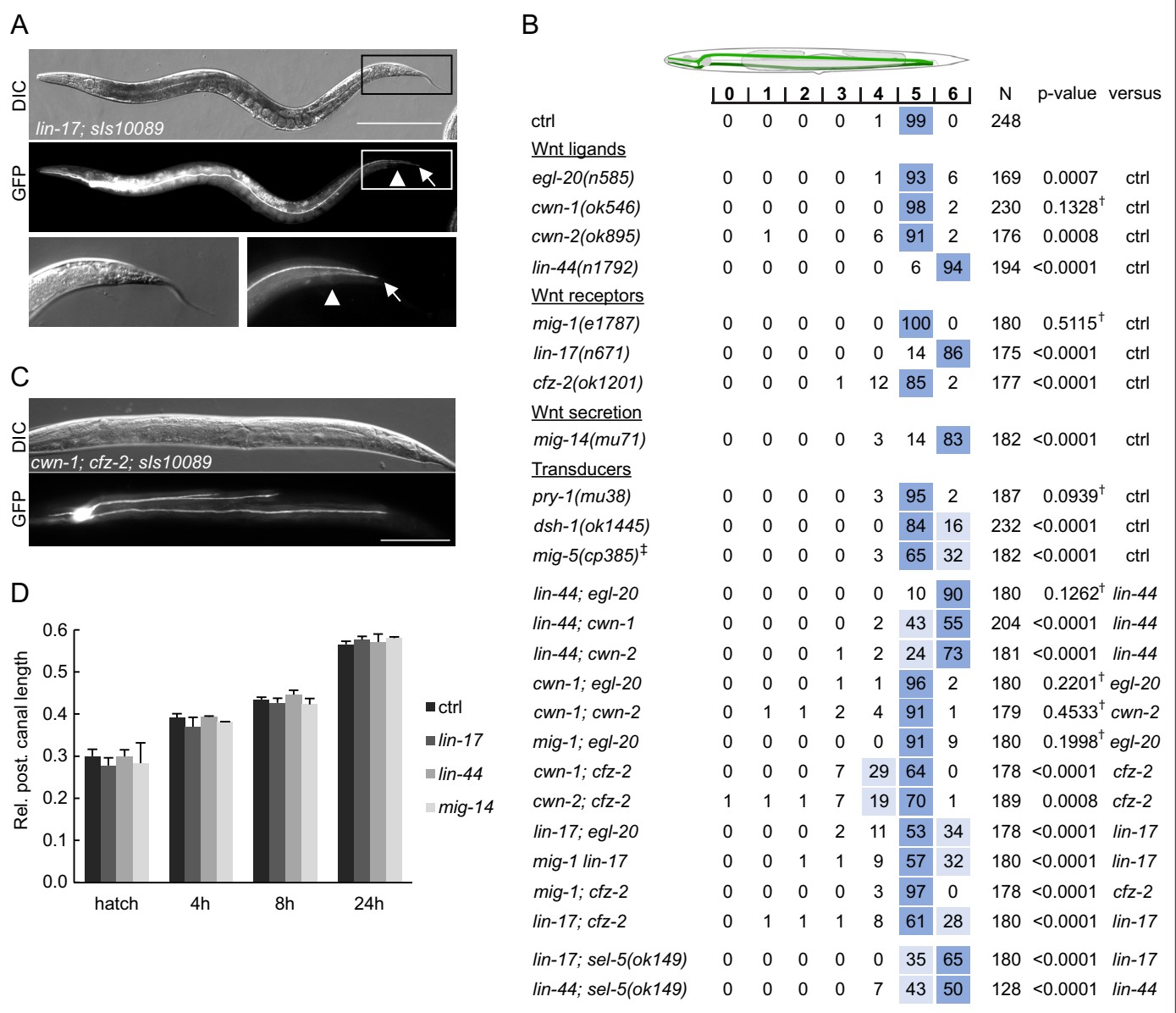

**Figure 7.** Wnt pathway components determine the length of the posterior excretory canal. (**A**) Posterior canals of the excretory cell overgrow into the tip of the tail in *lin-17* mutants. The excretory cell was visualized by *Ppgp-12::gfp* expression from the *sIs10089* transgene. Boxed areas are magnified in the bottom row, scale bare represents 200 µm. (**B**) Quantification of posterior canal outgrowth in Wnt pathway mutants. Canal outgrowth was scored and statistically evaluated as in *Figure 6B* and a comparison was made either to the control strain (*sIs10089* only) or to the more severe single mutant in the case of double mutant strains. All strains contained the *sIs10089* transgene to visualize excretory canals except for *lin-17; sel-5* and *lin-44; sel-5* which contained *mamEx11*, and *mig-5*, which contained *mamEx29*. †Differences not significant. ‡ Full genotype was *mig-5(cp385[mNG-GLO^AID::mig-5])*; *mamEx29; mamEx34[eft-3p::TIR::mRuby; myo2p::tdTomato]* and animals were grown on NGM plates with 1 mM auxin from L1 till L4/young adult stage. Only animals containing the *mamEx34* array were used for analysis. (**C**) Three branches of the posterior canal were detected in *cwn-1; cfz-2* animals. The excretory canal was visualized by the *sIs10089* transgene, the scale bar represents 200 µm. (**D**) Posterior excretory canals display normal length during early larval development. The dynamics of posterior canal outgrowth in *lin-44*, *lin-17*, and *mig-14* mutants was assessed by measuring the canal length and normalizing it to the total body length at hatching and at three time points during early larval development. Results are presented as mean + s.d. of at least 30 measured canals. Comparison between samples assessed by unpaired two-tailed Student's *t*-test did not reveal any significant differences.

the mutants but rather that these mutants miss a signal determining the final position of the canal in later larval stages. Simultaneous loss of *lin-17* and *sel-5* led to a partial rescue of the *lin-17* phenotype (*Figure 7B*). A similar effect was observed in *lin-44; sel-5* double mutants (*Figure 7B*). Analysis of a *lin-17; sel-5 vps-29* triple mutant was not possible as this combination was lethal. These data indicate that *sel-5* is required for the initial growth of the excretory canals. The LIN-44/Wnt signal acting through

the LIN-17/Frizzled receptor is either required early but the phenotypic manifestation is postponed or it is required at a later stage to terminate excretory canal extension.

## Discussion

The NAK family kinase AAK1 has recently been added to the long list of Wnt signalling regulators (*Agajanian et al., 2019*), acting at the level of LRP6 endocytosis. In the present study, we probed the role of the *C. elegans* AAK1 orthologue SEL-5 in Wnt signalling and uncovered its requirement for Wnt-dependent QL neuroblast daughter cell migration. We also revealed that SEL-5 is required during the outgrowth of the excretory canal cell prior to a Wnt-dependent stop signal. Importantly, in both instances the role of SEL-5 is independent of its kinase activity, defining an alternative mode of action for this kinase.

The role of SEL-5 has not been extensively studied in *C. elegans*; the only available reports uncovered the role of *sel-5* as a genetic suppressor of a constitutively active *lin-12*/Notch mutation (*Tax et al., 1997*; *Fares and Greenwald, 1999*). Information about protein function was missing. SEL-5 shows the highest similarity to the mammalian Numb-associated kinase family members AAK1 and BMP2K (*Shaye and Greenwald, 2011*; *Sorrell et al., 2016*). Both AAK1 and BMP2K were shown to phosphorylate the μ2 subunit of the AP2 adaptor at position T156 and thus regulate clathrin-mediated endocytosis (*Conner and Schmid, 2002*; *Ricotta et al., 2002*; *Ramesh et al., 2021*). Our findings uncovered some similarities between SEL-5 and AAK1 or BMP2K. Firstly, we show that the level of DPY-23/AP2M1 phosphorylation in *sel-5* mutants is decreased compared to controls. Secondly, the subcellular localization of SEL-5 to punctate structures proximal to the plasma membrane in oocytes and epidermal seam cells resembles the subcellular distribution of human AAK1 (*Conner and Schmid, 2002*). These data support the notion that SEL-5 has a function analogous to its mammalian counterparts.

Given the established role of AAK1 in the regulation of mammalian endocytosis (*Kadlecova et al., 2017*; *Agajanian et al., 2019*; *Wrobel et al., 2019*), it was striking to see that loss of *sel-5* expression alone had no obvious developmental consequences. However, even in mammalian cells complete loss of AP2M1 T156 phosphorylation does not block endocytosis but only reduces its efficiency (*Motley et al., 2006*; *Wrobel et al., 2019*). We did not observe complete loss, but only a partial reduction of DPY-23 phosphorylation in *sel-5* mutants. This implies that SEL-5 is not the only kinase responsible for DPY-23 phosphorylation. It also suggests that endocytosis is not inhibited but could be only partially compromised. In such case, the level of ongoing endocytosis might be sufficient to support proper developmental decisions, such as the direction of Q neuroblast migration. We argued that in combination with an additional insult to the intracellular trafficking machinery, in this case the retromer complex, the suboptimal function of endocytosis would manifest itself. Indeed, *sel-5 vps-29* double mutants display two highly penetrant phenotypes, the QL.d migration defect and shortening of the excretory cell canals. Strikingly, neither of these phenotypes can be attributed to a decrease in endocytic efficiency due to a lack of DPY-23 phosphorylation. Firstly, both phenotypes can be rescued by a tissue-specific expression of a kinase-inactive SEL-5 mutant protein. Secondly, *vps-29; dpy-23(mew25)* mutants expressing only non-phosphorylatable DPY-23 do not phenocopy *sel-5 vps-29* mutants. Alternative mechanisms of SEL-5 function thus have to be considered. It is possible that either SEL-5 affects endocytosis by a mechanism not dependent on DPY-23 phosphorylation or that SEL-5 regulates QL.d migration and excretory cell canal extension by endocytosis-independent mechanisms. Our findings about MIG-14/Wls behaviour, the most obvious endocytic cargo candidate for QL.d migration, argue against the first possibility; most notably the fact that MIG-14 does not re-localize to the plasma membrane in the absence of SEL-5. However, the effect of SEL-5 on a different endocytic cargo cannot be excluded. We detected significant shortening of the EGL-20 gradient in the *sel-5 vps-29* mutants. Given that SEL-5 is required both in the EGL-20-producing cells and in the muscle for QL.d migration, it is possible that some receptors that serve as a sink for EGL-20 are incorrectly accumulated at the plasma membrane and thus partially hinder EGL-20 spreading. In case of the excretory canal extension, SEL-5 could regulate the turnover of guidance receptors at the plasma membrane of the growing tip. In *Drosophila*, the loss of Numb-associated kinase (Nak) results in the absence or shortening of higher-order dendrites and Nak has been shown to promote endocytosis of Neuroglian/L1-CAM during dendrite extension and branching (*Yang et al., 2011*). In the *C. elegans* excretory cell, PAT-3/β1-integrin is required for canal outgrowth (*Hedgecock et al., 1987*). Integrins constantly

cycle between the plasma membrane and intracellular compartments using various trafficking routes (*Moreno-Layseca et al., 2019*) and could thus serve as a putative SEL-5 cargo.

A possible mechanism for how SEL-5 could affect endocytosis independently of DPY-23 phosphorylation follows from the finding that SEL-5 physically interacted with REPS-1 protein in a yeast-two-hybrid assay (*Tsushima et al., 2013*). REPS-1 is an orthologue of mammalian REPS1 (RALBP1-associated Eps Domain-containing 1), an Eps homology-containing protein known to interact with RALBP1 (RalA-binding protein 1) (*Yamaguchi et al., 1997*). RALBP1 is an effector of Ral GTPase that can interact with the μ2 subunit of AP2 and regulate endocytosis (*Jullien-Flores et al., 2000*). Importantly, human AAK1 interacted with both RALBP1 and REPS1 in several high-throughput mass spectrometry analyses (*Huttlin et al., 2017*; *Huttlin et al., 2021*; *Cho et al., 2022*; *Golkowski et al., 2023*). SEL-5 could thus regulate the AP2 complex indirectly via interaction with the REPS1-RALBP1 complex.

Searching for a possible endocytosis-independent mechanism of SEL-5 activity is complicated by the fact that while in the case of QL.d migration SEL-5 function is cell-nonautonomous, cell-autonomous expression of SEL-5 is sufficient to rescue the *sel-5 vps-29* excretory canal extension defect. This suggests that more than one mechanism might exist. One possibility common to both phenotypes comes from recent studies in mammalian cells. BMP2K has been shown to localize to the early secretory compartment and to regulate COPII coat assemblies (*Cendrowski et al., 2020*). SEL-5 could thus affect Wnt secretion in the case of QL.d migration or new guidance receptor delivery to the plasma membrane in the case of the excretory canals. EGL-20 gradient analyses revealed gradient shortening in *sel-5 vps-29* mutants. However, no significant differences in the number of EGL-20 puncta were detected between wild type and *sel*-5 animals, but also not between *vps-29* and *sel-5 vps-29*, despite the almost eightfold difference in the QL.d migration phenotype penetrance between these two mutants. This would argue against a role for SEL-5 in secretion. Shorter gradient in the double mutants could thus more likely arise from a combination of partially reduced EGL-20 secretion caused by *vps-29* and a partial reduction of EGL-20 spreading caused by a possible *sel-5*-dependent membrane retention of a receptor acting as an EGL-20 sink. A direct correlation between the amount of EGL-20, the range of the EGL-20 gradient, and the QL.d phenotype would be necessary to explain the phenotypic differences between the various mutants. More experiments will thus be required to establish how SEL-5 exerts its function. The biggest question to answer will be how it does so independently of its kinase activity.

Apart from the role of SEL-5 and the retromer complex in excretory cell canal extension, we have also uncovered a significant contribution of Wnt pathway components in defining the length of the excretory canal. The role of Wnt signalling in the outgrowth of the excretory canals has been suggested (*Sundaram and Buechner, 2016*); however, the only evidence came from the early observation that canals overgrow in *lin-17* mutants (*Hedgecock et al., 1987*) and that erroneous canal branching is

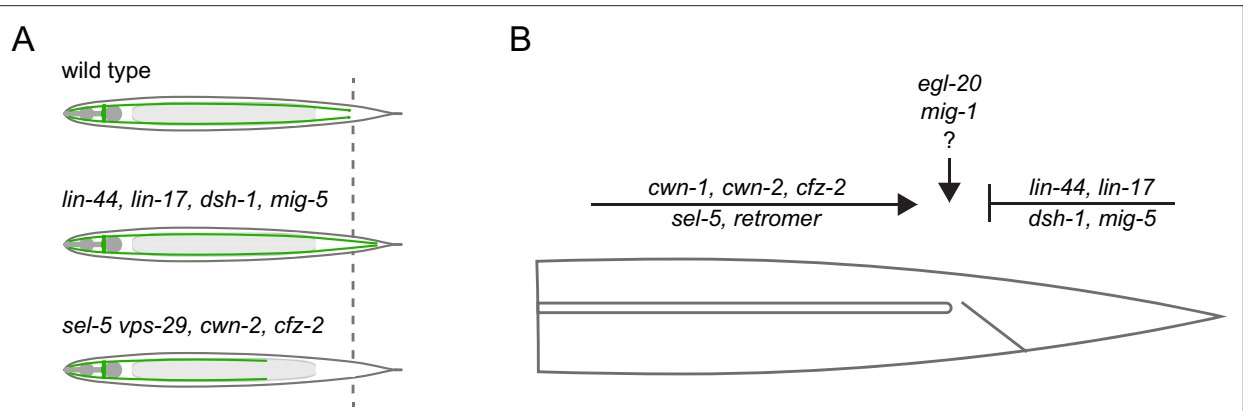

**Figure 8.** Model summarizing the inputs acting in the excretory cell outgrowth. (**A**) Three possible outcomes of the canal outgrowth depending on the genetic background. (**B**) Model summarizing the role of Wnt pathway components, *sel-5*, and the retromer in the outgrowth of the excretory cell posterior canals. Only players discussed in this work are depicted. First, the active outgrowth of the posterior canals is controlled by *sel-5*, retromer, and the Wnt pathway consisting of *cwn-1*, *cwn-2*, and *cfz-2*. It remains to be determined whether these players act sequentially, in parallel or as components of a single pathway. After reaching the proper length, further canal extension is inhibited by a second Wnt signalling pathway governed by *lin-44*, *lin-17*, *dsh-1*, and *mig-5*. Lastly, *egl-20* and *mig-1* might fine-tune the precise stopping position of the growing canal.

observed in *axl-1*/Axin-like mutants (*Oosterveen et al., 2007*). More recently, it has been shown that the loss of PLR-1, a transmembrane E3 ubiquitin ligase, results in excretory canal shortening (*Bhat et al., 2015*). PLR-1 is responsible for internalization of Frizzled receptors from the cell surface (*Moffat et al., 2014*). However, a direct link between PLR-1 function in the excretory cell and Wnt signalling has not been established. Our analysis of single Wnt and Frizzled mutants revealed that while loss of *cwn-2* or *cfz-2* expression resulted in a very mild shortening of the excretory canal, loss of *lin-44* or *lin-17* led to profound canal overgrowth (summarized in *Figure 8A*). These findings suggested that two independent Wnt pathways could be employed to establish a proper excretory canal length – one that promotes canal extension and one that generates the stop signal for growth termination. Further analyses of double mutants and other Wnt signalling components revealed that the extension-promoting pathway includes *cwn-1* in addition to *cwn-2* and *cfz-2*, while the stop-signal pathway encompasses *lin-44*, *lin-17*, *dsh-1*, *mig-5*, and *mig-14*. A similar repulsive role of LIN-44/LIN-17 complex has been described in the case of a posterior axon of the *C. elegans* GABAergic DD6 motor neuron (*Maro et al., 2009*) or PLM, ALN, and PLN neurons (*Zheng et al., 2015*). Loss of *lin-44* or *lin-17* expression promoted outgrowth of the posterior neurites of these neurons, implicating that in wild type animals, LIN-44 serves as a repulsive cue. On the other hand, *cwn-2* and *cfz-2* were shown to positively regulate the posterior neurite outgrowth of RMED/V neurons with *cwn-2* acting as an attractive cue (*Song et al., 2010*). The role of two other Wnt signalling components, *egl-20* and *mig-1*, is less clear. No effect (*mig-1*) or only a very mild overgrowth defect (*egl-20*) is observed in single mutants. However, both *egl-20* and *mig-1* significantly rescue the overgrowth phenotype of *lin-17* mutants, while at the same time, *mig-1* can suppress the shortening of canals in *cfz-2* mutants. EGL-20-producing cells are localized around the rectum (*Whangbo and Kenyon, 1999*; *Harterink et al., 2011a*), exactly where the excretory canals stop, while LIN-44 is expressed more posteriorly (*Herman et al., 1995*; *Harterink et al., 2011a*). A possible explanation could thus be that while LIN-44 provides a general posterior repulsive signal, EGL-20 fine-tunes the exact stopping position of the growing canal. The role of different Wnts and Frizzleds in excretory canal outgrowth is summarized in *Figure 8B*.

Further investigation will be required to decipher the exact way how SEL-5 and the retromer crosstalk with Wnt signalling during excretory cell outgrowth. It is clear though that more than one mechanism is likely involved. First, *sel-5 vps-29* mutants display canal shortening similarly to *cwn-1; cfz-2* or *cwn-2; cfz-2*, suggesting a positive regulatory role. Mutants in *lin-17* and *lin-44* display canal overgrowth, yet *sel-5* is partially able to suppress this phenotype. This would imply a negative regulatory role for *sel-5* and be in agreement with the role of AAK1 in Wnt pathway regulation (*Agajanian et al., 2019*). However, *sel-5* and *vps-29* are required already during the initial larval outgrowth while the LIN-44/LIN-17 signal is required later. The observed rescue might thus also be explained by delayed canal growth and not by a direct impact of *sel-5* and *vps-29* on LIN-44 or LIN-17 levels or localization. Alternatively, *sel-5* and *vps-29* may regulate LIN-44 or LIN-17 earlier but the signalling outcome does not manifest itself until later in development.

Taken together, we have uncovered cooperation between the *C. elegans* orthologue of the AAK1 kinase, SEL-5, and the retromer complex in regulating cell migration and cell outgrowth, and we have also outlined the engagement of Wnt pathway components in defining the length of the excretory cell canals. The surprising independence of the SEL-5 function on its kinase activity opens a new path for investigation with high relevance also for mammalian biology. AAK1 has been proposed as a potential drug target for treating various neurological disorders and preventing viral entry, where kinase activity is the target of inhibition (*Martinez-Gualda et al., 2022*; *Xin et al., 2023*). In light of our findings, the non-enzymatic activities of AAK1 should also be taken into account.

## Materials and methods

**Key resources table**

| Reagent type (species) or resource | Designation | Source or reference | Identifiers | Additional information |
|---|---|---|---|---|
| Strain, strain background (*Caenorhabditis elegans*) | Wild type | Caenorhabditis Genetics Center | N2 | Strain can be obtained from CGC |

*Continued on next page*

*Continued*

| Reagent type (species) or resource | Designation | Source or reference | Identifiers | Additional information |
|---|---|---|---|---|
| Strain, strain background (*C. elegans*) | *lin-17(n671) I* | CGC, *Brenner, 1974* | MT1306 | Strain can be obtained from CGC |
| Strain, strain background (*C. elegans*) | *lin-44(n1792) I* | CGC, *Herman and Horvitz, 1994* | MT5383 | Strain can be obtained from CGC |
| Strain, strain background (*C. elegans*) | *mig-1(e1787) I* | CGC, *Brenner, 1974* | CB3303 | Strain can be obtained from CGC |
| Strain, strain background (*C. elegans*) | *pry-1(mu38) I; him-5(e1490) V* | CGC, *Maloof et al., 1999* | CF491 | Strain can be obtained from CGC |
| Strain, strain background (*C. elegans*) | *cwn-1(ok546) II* | CGC | RB763 | Strain can be obtained from CGC |
| Strain, strain background (*C. elegans*) | *dsh-1(ok1445) II* | CGC | RB1328 | Strain can be obtained from CGC |
| Strain, strain background (*C. elegans*) | *mig-14(mu71) II* | CGC, *Harris et al., 1996* | CF367 | Strain can be obtained from CGC |
| Strain, strain background (*C. elegans*) | *mig-5(cp385[mNG-GLO^AID::mig-5]) II* | CGC, *Heppert et al., 2018* | LP728 | Strain can be obtained from CGC |
| Strain, strain background (*C. elegans*) | *vps-35(hu68) II* | CGC, *Coudreuse et al., 2006* | KN555 | Strain can be obtained from CGC |
| Strain, strain background (*C. elegans*) | *sel-5(ok149) III* | CGC | GS2381 | Strain can be obtained from CGC |
| Strain, strain background (*C. elegans*) | *sel-5(ok363) III* | CGC | RB638 | Strain can be obtained from CGC |
| Strain, strain background (*C. elegans*) | *egl-20(n585) IV* | CGC, *Harris et al., 1996* | MT1215 | Strain can be obtained from CGC |
| Strain, strain background (*C. elegans*) | *cwn-2(ok895) IV* | CGC | VC636 | Strain can be obtained from CGC |
| Strain, strain background (*C. elegans*) | *cfz-2(ok1201) V* | CGC | RB1162 | Strain can be obtained from CGC |
| Strain, strain background (*C. elegans*) | *muIs32[Pmec-7::gfp; lin-15(+)] II* | CGC, *Ch'ng et al., 2003* | CF702 | Strain can be obtained from CGC |
| Strain, strain background (*C. elegans*) | *huSi2[Pmig-14::mig-14::gfp] II* | *Lorenowicz et al., 2014* | KN1312 | Korswagen lab |
| Strain, strain background (*C. elegans*) | *dpy-5(e907) X; sIs10089 III* | CGC, *McKay et al., 2003* | BC10210 | Strain can be obtained from CGC |
| Strain, strain background (*C. elegans*) | *lin-15AB(n765) kyIs4 [Pceh-23::unc-76::gfp+lin-15(+)] X* | CGC, *Zallen et al., 1999* | CX2565 | Strain can be obtained from CGC |
| Strain, strain background (*C. elegans*) | *fcho-1(ox477) II; dpy-23(mew25) X* | Gunther Hollopeter | GUN27 | Hollopeter lab |
| Strain, strain background (*C. elegans*) | *vps-29(tm1320)III; muIs32 [Pmec-7::gfp] II* | *Yang et al., 2008* | | Korswagen lab |
| Antibody | Anti-α-tubulin antibody (mouse monoclonal) | Sigma-Aldrich | RRID:AB_477593 | WB (1:10,000) |
| Antibody | Anti-GFP antibody (mouse monoclonal) | Roche | RRID:AB_390913 | WB (1:4000) |
| Antibody | Anti-phosphoAP2M1 (rabbit monoclonal) | Abcam | RRID:AB_10866362 | WB (1:2500) |
| Antibody | Goat anti-mouse HRP-conjugated antibody (goat polyclonal) | Jackson ImmunoResearch Laboratories | RRID:AB_2307392 | WB (1:10,000) |

*Continued on next page*

*Continued*

| Reagent type (species) or resource | Designation | Source or reference | Identifiers | Additional information |
|---|---|---|---|---|
| Antibody | Goat anti-rabbit HRP-conjugated antibody (goat polyclonal) | Jackson ImmunoResearch Laboratories | RRID:AB_2337938 | WB (1:10,000) |
| Recombinant DNA reagent | pDD282 (plasmid) | *Dickinson et al., 2013* | RRID:Addgene_66823 | |
| Recombinant DNA reagent | pDD162 (plasmid) | *Dickinson et al., 2013* | RRID:Addgene_47549 | |
| Recombinant DNA reagent | pCFJ104 (plasmid) | *Frøkjaer-Jensen et al., 2008* | RRID:Addgene_19328 | |
| Recombinant DNA reagent | pPD95.81 (plasmid) | Addgene (Andrew Fire) | RRID:Addgene_1497 | |
| Software, algorithm | Fiji image processing package | *Schindelin et al., 2012* | | |
| Software, algorithm | Real Statistics Resource Pack, release 7.6 | *Zaiontz, 2021*, https://www.real-statistics.com | | |
| Software, algorithm | MATLAB | https://www.mathworks.com/ | | |

## *Caenorhabditis elegans* strains and culture

Standard methods of cultivation, manipulation, and genetics of *C. elegans* were used as described previously (*Brenner, 1974*). Bristol N2 strain was used as wild type, and *Escherichia coli* strain OP50 was used as a food source. Other strains, extrachromosomal, and integrated arrays used in this study were:

LGI: *pry-1(mu38)* (*Maloof et al., 1999*), *lin-17(n671)* (*Brenner, 1974*), *lin-44(n1792)* (*Herman and Horvitz, 1994*), *mig-1(e1787)* (*Brenner, 1974*).

LGII: *cwn-1(ok546)*, *dsh-1(ok1445)*, *mig-14(mu71)* (*Harris et al., 1996*), *muIs32* [*Pmec-7::gfp; lin-15(+)*] (*Ch'ng et al., 2003*), *huSi2* [*Pmig-14::mig-14::gfp*], *mig-5(cp385[mNG-GLO^AID::mig-5])* (*Heppert et al., 2018*), *vps-35(hu68)* (*Coudreuse et al., 2006*)

LGIII: *sel-5(ok363)*, *sel-5(ok149)*, *vps-29(tm1320)*, *sls10089* (*McKay et al., 2003*).

LGIV: *egl-20(n585)* (*Harris et al., 1996*), *cwn-2(ok895)*.

LGV: *muIs35* [*Pmec-7::gfp; lin-15(+)*] (*Ch'ng et al., 2003*), *cfz-2(ok1201)*.

LGX: *dpy-23(mew25)* (G. Hollopeter and G. Beacham, unpublished), *kyIs4* [*Pceh-23::unc-76::gfp+lin-15(+)*] (*Zallen et al., 1999*).

Full list of strains generated in this study is provided in *Supplementary file 1*.

## Molecular biology, germline transformation, and RNA interference

Total RNA was isolated using the TRIzol reagent (Cat# 15596026, Thermo Fisher Scientific) from a mixed-stage population of *C. elegans* N2 strain collected from a single 9 cm NGM plate. cDNA was transcribed from total RNA using SuperScript III Reverse Transcriptase (Cat# 18080093, Thermo Fisher Scientific) and an oligo(dT) primer. Full-length *sel-5* (isoform a), *vps-29* (isoform a), or *dpy-23* (isoform b) cDNA was PCR-amplified using gene-specific primers. PCR products were cloned into pJet1.2/blunt vector (Cat# K1232, Thermo Fisher Scientific) and sequenced. The same procedure was applied to clone truncated *sel-5* cDNA from *sel-5(ok363)* and *sel-5(ok149)* mutant animals. Two- or three-fragment Gibson assembly (*Gibson et al., 2009*) was employed to construct tissue-specific gene expression vectors. For that, 524 bp of *col-10* promoter sequence, 600 bp of *eft-3* promoter sequence, 4419 bp of *egl-20* promoter sequence, 2896 bp of *hlh-1* promoter sequence, 3410 bp of *pgp-12* promoter sequence, and 1629 bp of *wrt-2* promoter sequence were PCR-amplified from N2 genomic DNA, cDNAs were amplified from the corresponding pJet1.2 vectors and pPD95.81 (a gift from Andrew Fire [Addgene plasmid # 1497; http://n2t.net/addgene:1497; RRID:Addgene_1497]) was a source of backbone and GFP sequences. Fragments were assembled in the desired combinations and correct assembly was verified by sequencing. SEL-5 was tagged with GFP at its N-terminus, VPS-29 and DPY-23 were tagged at their C-termini. Point mutations were introduced into *sel-5* and *dpy-23* sequences using the QuikChange Site-directed Mutagenesis kit (Agilent). Transcriptional reporter P*pgp-12::gfp* was created by inserting 3410 bp of *pgp-12* promoter upstream of GFP in pPD95.81. P*egl-20::mCherry::PH* was constructed by Gibson assembly using P*wrt-2::gfp::PH* (*Wildwater et al., 2011*) and pCFJ104 (*Frøkjaer-Jensen*

*et al., 2008*; a gift from Erik Jorgensen [Addgene plasmid # 19328; http://n2t.net/addgene: 19328; RRID:Addgene_19328]) as PCR templates to amplify PH domain and mCherry, respectively. Sequences of all primers used for plasmid vector construction are listed in *Supplementary file 2*. To create extrachromosomal arrays, constructs were microinjected into distal gonads of young adults using Leica DMi8 inverted microscope equipped with DIC filters and InjectMan 4 and FemtoJet 4i microinjection system (Eppendorf). Microinjection mixtures contained 10 ng/µL of the plasmid of interest, 5 ng/µL P*myo-2::tdTomato* as a co-injection marker, and 135 ng/µL pBluescript as a carrier DNA.

CRISPR/Cas9 SEC knock-in method (*Dickinson et al., 2015*) was used to generate GFP-tagged endogenous *sel-5* locus and performed virtually as described (*Dickinson et al., 2015*; http://worm-cas9hr.weebly.com/protocols.html). 647 bp upstream of *sel-5* ATG (5' homology arm) and 558 bp *sel-5* genomic sequence starting from ATG (3' homology arm) were PCR-amplified from N2 genomic DNA with primers harbouring overhangs for Gibson assembly with the pDD282 vector (*Dickinson et al., 2013*; a gift from Bob Goldstein [Addgene plasmid # 66823; http://n2t.net/addgene:66823; RRID:Addgene_66823]) and assembled with pDD282 digested with ClaI and SpeI. Sequence GCTGAAAA GCCCTAGAGGCA was inserted into pDD162 vector for sgRNA expression (*Dickinson et al., 2013*; pDD162 was a gift from Bob Goldstein [Addgene plasmid # 47549; http://n2t.net/addgene:47549; RRID:Addgene_47549]). Sequences of all primers used for gene editing vector construction are listed in *Supplementary file 2*. Injection was performed as described above, injection mix contained 10 ng/µL pDD282 with homology arms, 50 ng/µL pDD162 with corresponding sgRNA sequence, and 5 ng/µL P*myo-2::tdTomato*. Line expressing P*sel-5::gfp* was first established, and several L1 animals were then exposed to a heat shock for 4 hr at 34°C to excise the selection SEC cassette and create a P*sel-5::gfp::sel-5* knock-in line. The EGL-20::GFP knock-in strain was generated as described (*Pani and Goldstein, 2018*) with one modification – the pDD282 vector was used for cloning the homology arms.

RNA interference experiments were conducted by feeding using bacterial strains from the Ahringer library (*Kamath and Ahringer, 2003*) except for *dpy-23*, which came from the Vidal library (*Rual et al., 2004*). L4 larvae were transferred to RNAi plates with the desired bacterial clones and the effect of RNAi was assessed in the next generation after 3–4 days at 20°C.

## Protein isolation and western blotting

For MIG-14::GFP detection, gravid hermaphrodites were subjected to hypochlorite treatment and released embryos were left to hatch overnight in M9 buffer. Larvae were collected and washed twice in M9 buffer. Pelleted larvae were then resuspended in TX-114 buffer (25 mM Tris–HCl pH 7.5, 150 mM NaCl, 0.5 mM CaCl$_2$, 1% TX-114, and cOmplete protease inhibitors [Roche]), snap-frozen, and then ground in liquid nitrogen. Thawed lysates were centrifuged at 20,800 × *g* for 30 min at 4°C, mixed with Laemmli sample buffer, separated on 8% SDS-PAGE, and transferred onto Amersham Protran Premium western blotting nitrocellulose membrane. GFP was detected with monoclonal anti-GFP antibody (Cat# 11814460001, Roche, RRID:AB_390913) and equal loading was assessed by staining with monoclonal anti-α-tubulin antibody (Cat# T9026, Sigma-Aldrich, RRID:AB_477593). Secondary goat anti-mouse HRP-conjugated antibody (Cat# 115-035-146, Jackson ImmunoResearch Laboratories, RRID:AB_2307392) and WesternBright ECL HRP substrate (Cat# K-12045-D20, Advansta) were used to visualize the signal. Images of membranes were taken on ImageQuant (LAS4000) and the Fiji Gels plug-in was used for subsequent densitometric analysis. For endogenous phosphorylated DPY-23 and DPY-23::GFP detection, 200 L4 larvae or young adults positive for DPY-23::GFP were collected into M9 buffer for each analysed strain. Animals were washed once with M9 buffer and two times with M9 buffer with 0.001% Triton X-100. 33 µL of animal pellet in M9/0.001% Triton X-100 was mixed with 33 µL of 4× Laemmli buffer, 4 µL of 20× PhosStop phosphatase inhibitors (Roche), 4 µL of 20× cOmplete protease inhibitors (Roche), and 1.6 µL of 1 M DTT. Samples were snap-frozen in liquid nitrogen, thawed on ice, and sonicated with 2 × 10 pulses at 0.8 amplitude, 0.85 duty cycle with UP50H ultrasound processor (Hielscher). Samples were heated to 99°C for 5 min and centrifuged at 20,000 × *g* for 10 min. 20 µL of each sample were loaded on 9% SDS-PAGE, separated and further processed as above using anti-GFP and anti-phosphoAP2M1 (Cat# ab109397, Abcam, RRID:AB_10866362) antibodies. Goat anti-rabbit HRP-conjugated antibody (Cat# 111-035-045, Jackson ImmunoResearch Laboratories, RRID:AB_2337938) was used to detect anti-phosphoAP2M1 signal.

## C. elegans phenotypes, microscopy, and statistical analyses

The final position of QL.paa (PVM – QL.d phenotype) and polarity of ALM and PLM neurons were assessed in L4 larvae carrying transgene *muIs32* or *muIs35*. PVM migration was scored as defective when PVM was located anteriorly to the posterior edge of the vulva. ALM polarity was scored as defective when bipolar or reversed neurites were observed. CAN neurons were visualized with *kyIs4* transgene and their position relative to V3 seam cell was scored. CAN positioned anterior to V3 was scored as displaced. For DiI staining, L3-L4 well-fed animals were washed from a plate with M9 buffer and incubated for 3 hr in 10 µg/mL DiI solution (D282, Thermo Fisher Scientific, dissolved to 2 mg/mL stock solution in dimethylformamide). Animals were then washed three times in M9 buffer and directly observed. Excretory cell canal lengths were scored in L4 or young adult animals carrying integrated P*pgp-12::gfp* transgene (*sIs10089*) or P*pgp-12::gfp* expressed from an extrachromosomal array (*mamEx11 or mamEx29*). Posterior canals were scored as wild type when they reached the region between the inner edge of the posterior gonadal turn and the rectum. Shortened canal phenotype was graded according to the region within the worm body the canal reached. The following landmarks were used for each category: 0 – excretory canal missing entirely or reaching the inner edge of anterior gonadal turn; 1 – canal reaching anterior spermatheca; 2 – canal reaching vulva; 3 – canal reaching posterior spermatheca; 4 – canal reaching inner edge of posterior gonadal turn; and 5 – canal reaching rectum (wild type). Canals that overgrew the rectum and reached the tail tip were scored as 6. Anterior excretory cell canals were measured and their length normalized to the measured distance between the posterior edge of the pharynx and the nose tip. For the assessment of posterior canal extension dynamics, larval excretory cell posterior canals were measured at four developmental stages (hatch, 4, 8, and 24 hr after hatching) and normalized to the animal body length. For microscopy imaging, worms were anaesthetized either with 10 mM sodium azide or 1 mM levamisol and mounted on 3% agarose pads. Images were taken on Leica DM6 upright microscope. For confocal imaging, Zeiss LSM880 confocal microscope was used. The EGL-20/Wnt gradient was visualized in L2/L3 animals using a spinning disc microscopy setup, based on a Nikon Eclipse Ti2 equipped with a Yokogawa CSU-W1 scan head, 50 um pinhole disc, 488 nm laser, Teledyne Photometrics PRIME BSI camera, ×60 W NA1.2 objective, and operated with NIS Elements. For each animal, a z-stack of 31 slices, dz = 1 µm was acquired. Imaging was done at room temperature (22–23°C). Images were processed with the Fiji image processing package (*Schindelin et al., 2012*). For EGL-20 gradient quantification, a maximum projection of seven consecutive z-sections where the rectal epithelial cells were visible was made for each sample. A curved line of 150 µm in length and 3.9 µm in width was fitted through the ventral side of each animal using the rectum as the starting point and the selected region was straightened using the 'Straighten' function in Fiji. To count the number of EGL-20 puncta in these selections, images were blinded and the puncta were manually counted. Counting was repeated three times and the values for each image were averaged. These averaged values were then plotted and used for statistical evaluation. For mean pixel intensity calculations, a Gaussian blur (sigma = 3) was applied to the straightened images. Subsequently, a single segmentation threshold was determined from a representative wild type image by using the 'graythresh' inbuilt function in MATLAB (which uses the Otsu method). This threshold was multiplied by 1.7 (a factor that was determined manually) to generate a mask that segments only bright EGL-20 spots. Using the derived threshold, a binary mask was made for all images. Using this mask, the mean pixel intensity of all pixels inside the segmented EGL-20 spots was computed. Statistical analyses were performed either in GraphPad Prism or in Excel equipped with the Real Statistics Resource Pack software (Release 7.6; *Zaiontz, 2021*).

## Materials availability

Strains and plasmids generated in this study are available upon reasonable request from academic researchers.

## Acknowledgements

We thank Gunther Hollopeter and Gwendolyn Beacham (Cornell University, USA) for sharing the dpy-23(mew25) allele. We thank Jan Mašek for critically reading the manuscript and Gunther Hollopeter for helpful discussion. Some strains were provided by the Caenorhabditis Genetics Center (CGC) which is supported by the National Institutes of Health – Office of Research Infrastructure Programs (P40 OD010440). This work was funded by Charles University Grant Agency grant 1446218/2018 to

FK, Czech Science Foundation grant 16-17966Y to MM, Lumina quaeruntur grant LQ200522301 of the Czech Academy of Sciences to TCM, and Charles University programme SVV 260559. Microscopy was performed in the Vinicna Microscopy Core Facility co-financed by the Czech-BioImaging large RI project LM2023050. The authors acknowledge the Imaging Methods Core Facility at BIOCEV, institution supported by the MEYS CR (LM2023050 Czech-BioImaging), for their support and assistance on the spinning disc imaging done in this work. Computational resources were supplied by the project 'e-Infrastruktura CZ' (e-INFRA LM2018140) provided within the program Projects of Large Research, Development and Innovations Infrastructures.

## Additional information

### Funding

| Funder | Grant reference number | Author |
|---|---|---|
| Grantová Agentura České Republiky | 16-17966Y | Marie Macůrková |
| Grantová Agentura, Univerzita Karlova | 1446218/2018 | Filip Knop |
| Akademie Věd České Republiky | Lumina quaeruntur grant LQ200522301 | Teije Corneel Middelkoop |

The funders had no role in study design, data collection and interpretation, or the decision to submit the work for publication.

### Author contributions

Filip Knop, Conceptualization, Formal analysis, Funding acquisition, Validation, Investigation, Visualization, Writing - original draft, Writing – review and editing; Apolena Zounarová, Conceptualization, Formal analysis, Validation, Investigation, Visualization, Writing – review and editing; Vojtěch Šabata, Investigation, Writing – review and editing; Teije Corneel Middelkoop, Formal analysis, Investigation, Writing – review and editing; Marie Macůrková, Conceptualization, Formal analysis, Supervision, Funding acquisition, Investigation, Visualization, Writing - original draft, Project administration, Writing – review and editing

### Author ORCIDs

Marie Macůrková (iD) https://orcid.org/0000-0002-7082-0093

### Decision letter and Author response

Decision letter https://doi.org/10.7554/eLife.91054.sa1
Author response https://doi.org/10.7554/eLife.91054.sa2

## Additional files

### Supplementary files

- Supplementary file 1. List of all *C. elegans* strains generated.
- Supplementary file 2. List of primers used to construct plasmid vectors.
- MDAR checklist

### Data availability

All data generated or analysed during this study are included in the manuscript and supporting files. Source data files have been provided for Figures 1, 3 and 5.

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
