## [Editor Report]

This important study defines developmental roles for a protein kinase involved in endocytosis and reports a surprising finding that the kinase catalytic activity is unnecessary. The evidence supporting the claims of the authors is solid. As this kinase was previously suggested to be a target of drug design efforts, this work will be of broad interest to cell biologists and biochemists.

---

## [Decision Letter]

**Decision letter after peer review:**

Thank you for submitting your article "*Caenorhabditis elegans* SEL-5/AAK1 regulates cell migration and cell outgrowth independently of its kinase activity" for consideration by *eLife*. Your article has been reviewed by 2 peer reviewers, and the evaluation has been overseen by a Reviewing Editor and Piali Sengupta as the Senior Editor.

Essential revisions (for the authors):

1. The authors propose a kinase-activity independent mode of action for SEL-5/AAK1, however, the evidence supporting this claim is inadequate, as it relies on under-characterized mutants for dpy-23 and sel-5. The claims will be stronger if the authors test for DPY-23 phosphorylation in the new dpy-23(mew25) mutant and further characterize whether mew25 behaves as a loss-of-function.

2. The authors need to have higher magnification/resolution images and use markers (e.g. for the plasma membrane, junctions, endosomes, golgi, etc.) to carefully analyze the localization of MIG-14::GFP in control vs. mutants.

3. Reviewers had some issues with the interpretation of data related to the sel-5 mutant phenotypes in EXC, and the genetic interactions with Wnt signaling mutants (e.g., egl-20, lin-44). Further genetic interactions (e.g., egl-20 and/or mig-1 mutations with lin-17) would strengthen this study.

4. There is no clear connection between mig-10 and Wnt signaling, and thus the inclusion of mig-10 in this report is not well supported nor necessary.

5. Some changes to statistical analyses are needed (see specific comments by reviewer 2).

6. The study lacks coherence and careful copy-editing is needed to improve the flow and quality of the text. For example, there is a clear hypothesis testing a role for SEL-5/AAK1 in DPY-23/AP2M1 phosphorylation and Wnt signaling. However, this model appears to be refuted but the authors do not mention/explore alternative targets or functions for SEL-5/AAK1.

*Reviewer #1 (Recommendations for the authors):*

1. The new dpy-23(mew25) allele requires further genetic characterization, such as the description of dominance/recessiveness, ability to complement existing dpy-23 mutants, description of any dpy-23 like phenotypes, ability to suppress/enhance other dpy-23-mediated processes, etc. Without knowing more about this allele, we cannot reliably interpret the lack of effects seen with this mutant.

2. I do not agree that, at least from the data shown, "neither levels nor subcellular distribution of a functional MIG-14::GFP protein changed significantly in sel-5 vps-29 compared to vps-29 single mutant". The images shown in Figure 3C are not sufficient to reach this conclusion and I could argue that I see more MIG-14::GFP at the apical and lateral membranes in sel-5 mutants than in control. To make a strong conclusion, the authors need to have higher magnification/resolution images and use markers (e.g. for plasma membrane, junctions, endosomes, golgi, etc.) to carefully analyze the localization of MIG-14::GFP in control vs. mutants. Similarly, the conclusion that "Levels of MIG-14::GFP in sel-5 single mutants were variable yet on average comparable to wild type" is not supported by Figure 3D, as a statistical analysis of this result is not presented, and it is possible that the high variability makes this result non-significant.

3. I am surprised that the authors reported results from only one multi-copy extrachromosomal array for their studies of SEL-5 kinase mutants. Given the variability in expression from such arrays, typically results are reported for multiple transgenes. More importantly, given that extrachromosomal arrays are known to highly overexpress the protein of interest, and that even proteins with reduced function can rescue when highly overexpressed, the conclusion that SEL-5 kinase activity is not required for function is not supported. These experiments should be done with single-copy insertion transgenes, or by introducing the desired mutations into the endogenous sel-5 locus. In addition, as described for dpy-23(mew25), further genetic characterization of sel-5 kinase mutants is warranted: do they cause phenotypes on their own? Do they affect other sel-5-mediated processes (i.e., look for genetic interactions with lin-12/Notch mutants)? Finally, since the sel-5 coding sequence is tagged with GFP, it would be important to know whether the mutations introduced affect SEL-5 accumulation, localization, etc. Again, this would be better done with either single-copy insertion transgenes or using the endogenous GFP-tagged sel-5.

4. Although the excretory canal outgrowth phenotypes caused sel-5, retromer, and some components of the Wnt signaling, are novel, this story is underdeveloped and lacks mechanistic insight. Moreover, some significant genetic interactions are not considered in the model. Specifically:

4a. egl-20/Wnt mutants have a mild, but significant, overgrowth defect, which suggests that EGL-20 acts in parallel to LIN-44 as a "stop" signal (more on this below). This is not mentioned.

4b. mig-1/Fzd mutants do not display an excretory canal outgrowth phenotype, however mig-1 loss significantly suppressed the reduced outgrowth caused by loss of cfz-2/Fzd. Moreover, mig-1 did not synergize with egl-20. These results, which are not further discussed, are consistent with a model where egl-20/Wnt and mig-1/Fzd act in a pathway (in parallel to lin-44/Wnt and lin-17/Fzd) as a "stop" signal, and also consistent with a model where even mild disruption of the "stop" signal can overcome defects caused by loss of a "go" signal mediated by cwn-2/Wnt and cfz-2/Fzd. Further genetic interactions (e.g., egl-20 and/or mig-1 mutations with lin-17, which displays a less penetrant phenotype than lin-44) would strengthen this model.

5. Implicating EGL-20/Wnt as part of the "stop" signal would allow the authors to use previously-published endogenously-tagged EGL-20 reporters (Pani and Goldstein, 2018) to assess whether the non-autonomous function of sel-5 in QL.d migration and excretory canal outgrowth is related to Wnt secretion, which would provide some mechanistic insight into SEL-5 function. In this regard, I also note that there is a transgenic CWN-2::Venus reporter that could be used to visualize CWN-2/Wnt accumulation in sel-5 mutants (rescue of the cwn-2 excretory canal defect by this transgene should be tested before using).

6. Focusing on mig-10 here is confusing. I do not agree that "So far, the most studied pathway regulating excretory canal outgrowth is the one involving UNC-53 […], ABI-1 […], and MIG-10", as many other pathways have been implicated in excretory canal outgrowth and guidance (i.e., Slit/Robo, FGFR, Rho/Rac signaling, the Par polarity pathway, etc.). The authors mention that thus far ligands/receptors that might regulate a UNC-53/ABI-1/MIG-10 are not known, thus perhaps they're suggesting that Wnt signaling would play this role. However, no work is presented to support this model. Thus, as it stands, there is no clear connection between mig-10 and Wnt signaling, and thus the inclusion of mig-10 in this report is not well supported nor necessary.

*Reviewer #2 (Recommendations for the authors):*

1. Please understand that this comment is communicated with the aim of improving the manuscript. I believe the writing quality can be improved with copy-editing. I encourage you to ask a copyeditor to scan the entire document for small errors. Below is one example of the kind of edits needed throughout the manuscript. (My additions are shown in brackets.)

From p. 15, line 372:

"[A] possible mechanism [for] how SEL-5 could affect endocytosis…"

2. In looking up the cited reference for plr-1 mutants, I noticed that the in-text citation misspells the first author's name. (The name is Bhat, rather than Bhatt.) It's possible this error could be caused by a reference manager software, or it is a typo. It would be useful to spell-check all in-text references.

[Editors' note: further revisions were suggested prior to acceptance, as described below.]

Thank you for resubmitting your work entitled "*Caenorhabditis elegans* SEL-5/AAK1 regulates cell migration and cell outgrowth independently of its kinase activity" for further consideration by *eLife*. Your revised article has been evaluated by a Senior Editor and a Reviewing Editor.

The manuscript has been improved but there are some remaining issues that need to be addressed, as outlined below:

1. The number or intensity of EGL-20 puncta on their own does not provide an accurate description of a morphogenetic gradient. The authors are encouraged to revise the text and/or quantify EGL-20 puncta over distance.

2. The evidence implicating sel-5 in only one pathway is not strong. The authors are encouraged to edit the text.

3. Both reviewers provide suggestions on how to improve the text. Copy-editing is needed, the EGL-20 gradient should not go under the section titled "SEL-5 does not affect MIG-14 endocytosis". Instead a new section titled "SEL-5 affects the EGL-20 gradient".

*Reviewer #1 (Recommendations for the authors):*

The authors have done an excellent job addressing the essential concerns raised in the first round of review with some great new experiments and clarifying changes to the text. In particular, the detailed and complete analysis of excretory canal defects in Wnt pathway mutants allowed the authors to propose an intriguing model describing a novel role for Wnt signaling in canal outgrowth. I also very much appreciate the more detailed analysis of the dpy-23(mew25) allele, which I think lends stronger support to their conclusion that DPY-23/AP2 phosphorylation by SEL-5/AAK1 is not the mechanism by which sel-5 regulates Wnt signaling.

However, I do have some concerns on the interpretation of new data re: the EGL-20/Wnt gradient presented in Figure 3, Supplement 2 (In general, I will note that the figures were not labelled in the manuscript; so, it was quite difficult to know what Figure I was looking at when reading the paper). The authors use two criteria to measure the EGL-20 gradient: number of EGL-20 puncta (Figure 3, Supp. 2, C) and signal intensity (Figure 3, Supp. 2, D), and found that neither sel-5 or vps-29 mutations (they do not specify which alleles were used for this experiment, at least I could not find it in the main text, figure legend, methods, or strain table) on their own significantly decreased the number or intensity of particles seen, while the double mutant had significantly fewer particles.

However, I don't think that the number or intensity of EGL-20 puncta on their own provides an accurate description of a gradient. A gradient is not only defined by the total amount of morphogen, but also how the concentration of this morphogen is distributed over distance, and the authors do not include any distance measures in their quantifications. I would argue, based on the representative images shown, that the EGL-20 gradient is indeed quite reduced in sel-5 and vps-29 mutants, because the extent of EGL-20 puncta in these mutants seems to be much shorter than in wildtype. Maybe a good measurement to include is distance from the rectum to the furthest anterior brightest puncta? In conclusion, based on the pictures shown, I agree that sel-5 is likely not significantly affecting the total amount of EGL-20 secreted, but I do not agree that the gradient (which should include a measure of the extent of spread) is unchanged.

I also don't think that the EGL-20 gradient should go under the section titled "SEL-5 does not affect MIG-14 endocytosis". Instead a new section titled "SEL-5 affects the EGL-20 gradient" (or "does not affect" depending on how the new measurements of gradient that incorporate distance turn out) should be added, and these figures should be part of the main text.

*Reviewer #2 (Recommendations for the authors):*

Overall, I was satisfied by most changes in the revised manuscript, "*Caenorhabditis elegans* SEL-5/AAK1 regulates cell migration and cell outgrowth independently of its kinase activity."

Weaknesses

1. Reviewer #2, comment 2. With the added data now in figure 6, I agree with the author's model that cwn-1, cwn-2, and cfz-2 act to promote EXC outgrowth while lin-44, lin-17, dsh-1, and mig-5 act as a stop signal.

However, I felt the evidence implicating sel-5 in only one pathway was not strong. On one hand, the observation that sel-5 vps-29 mutants had the same phenotype as cwn-1 cwn-2 mutants was consistent with it acting in the cwn-1/2 pathway. This claim was reasonable. The evidence that sel-5 did not influence the lin-44/lin-17 pathway was less clear. To support their claim, the authors argued that lin-44/lin-17 activity was not required early in development because lin-44/lin-17 mutants did not show a phenotype early. I would remind them that it is possible for gene activity to be required early and yet the mutant not show a phenotype until later. One would need a conditional allele to test the timing of requirement. (I am not suggesting this experiment!) I would advise them to alter the text in two places:

Line 382 LIN-17/Frizzled receptor is required at a later stage

Line 509 LIN-17 signal is required later

I would replace these claims with "loss of lin-17 does not result in a phenotype until a later stage."

Perhaps they should soften their claim that sel-5 does not regulate lin-44/lin-17. (The model is alright. It's just that -a model!)

2. Reviewer #2. Copy-editing was needed. There were a large number of syntax errors, particularly missing articles, prepositions, and punctuation. This was true in the introduction, results, and discussion.

---

## [Author Response]

Essential revisions (for the authors):1. The authors propose a kinase-activity independent mode of action for SEL-5/AAK1, however, the evidence supporting this claim is inadequate, as it relies on under-characterized mutants for dpy-23 and sel-5. The claims will be stronger if the authors test for DPY-23 phosphorylation in the new dpy-23(mew25) mutant and further characterize whether mew25 behaves as a loss-of-function.

The sel-5(ok149) allele was previously characterized by Fares and Greenwald (1999) as a loss-of-function allele. Our characterization of both sel-5(ok149) and sel-5(ok363) regarding the Wnt-related phenotypes is summarized in Table 1 and Figures 1B, 1D. We agree that the mew25 allele we use has not been extensively characterized before; however, mew25 and several other alleles introducing the same amino acid change were described previously by other groups (Hollopeter et al., 2014; Beacham et al., 2018; Partlow et al., 2019; Joseph et al., 2020). We have now made this clear in the text and we have included characterization of the allele (Figure 4). We show that no T160 phosphorylation is present in dpy-23(mew25) animals and we show that mew25 rescues the “jowls” phenotype of fcho-1 mutants, likely by destabilizing the closed conformation of AP2 (Hollopeter et al., 2014). Importantly, this allele is not a loss-of-function allele and does not show dpy-23 loss-of-function phenotypes. Because the T160A alleles of dpy-23 appear overtly wild-type, the mew25 allele (generated de novo in the N2 background via CRISPR) offered a unique opportunity to evaluate whether phosphorylation by SEL-5 is required for the phenotypes observed in our study. More details about in-text changes are provided in Response to Reviewer 1 – point 1, see below.

2. The authors need to have higher magnification/resolution images and use markers (e.g. for the plasma membrane, junctions, endosomes, golgi, etc.) to carefully analyze the localization of MIG-14::GFP in control vs. mutants.

We thank the reviewers for this feedback and agree that our statement about changes in MIG-14::GFP localization were not sufficiently supported. We have replaced the original images with confocal images acquired at higher magnification and we also included a panel comparing MIG-14::GFP together with an mCherry-PH membrane marker in all mutants (Figure 3, Figure 3 —figure supplement 1), which now supports our statement that in sel-5 vps-29 double mutants (or in either of the single mutants) MIG-14::GFP does not re-localize to the plasma membrane. This indicates that endocytosis of this cargo is not compromised. More details about in-text changes are provided in Response to Reviewer 1 – point 2, see below.

3. Reviewers had some issues with the interpretation of data related to the sel-5 mutant phenotypes in EXC, and the genetic interactions with Wnt signaling mutants (e.g., egl-20, lin-44). Further genetic interactions (e.g., egl-20 and/or mig-1 mutations with lin-17) would strengthen this study.

We highly appreciated this feedback as the incorporation of additional Wnt/Frizzled combinations in our analyses brought novel information about Wnt signals involved in excretory canal extension and helped to complement our model. We now show that cwn-1, cwn-2 and cfz-2 promote excretory canal extension, lin-44 and lin-17 prevent extension beyond the rectum, while egl-20 and mig-1 might help to define the exact stopping point for canal growth (Figure 6). More details about in-text changes are provided in Response to Reviewer 1 – point 4 and Reviewer 2 – point 2, see below.

4. There is no clear connection between mig-10 and Wnt signaling, and thus the inclusion of mig-10 in this report is not well supported nor necessary.

We agree that understanding the link between MIG-10 and SEL-5 would require more experiments and we have excluded it from the current manuscript.

5. Some changes to statistical analyses are needed (see specific comments by reviewer 2).

We thank Reviewer 2 for pointing out the usage of an inappropriate statistical test. We have corrected this mistake in all relevant Figures. Details about the changes are provided in Response to Reviewer 2 – point 1, see below.

6. The study lacks coherence and careful copy-editing is needed to improve the flow and quality of the text. For example, there is a clear hypothesis testing a role for SEL-5/AAK1 in DPY-23/AP2M1 phosphorylation and Wnt signaling. However, this model appears to be refuted but the authors do not mention/explore alternative targets or functions for SEL-5/AAK1.

We totally agree that deciphering the mechanism of SEL-5 action would move our study to a different level. We have now explored the possibility that SEL-5 activity is required for shaping of the Wnt gradient. To this end we have produced endogenously-tagged EGL-20::GFP line and assessed whether the amount of extracellular EGL-20 differs among the studied mutants (Figure 3 —figure supplement 2). However, our data do not support this possibility, leaving the mechanism of SEL-5 action still open. Recycling of membrane receptors, for example integrins, would be another possible scenario, but as we have no experimental evidence, we only briefly outline this possibility in the Discussion. We plan to explore this line in future but we feel it is beyond the scope of the current study.

Reviewer #1 (Recommendations for the authors):1. The new dpy-23(mew25) allele requires further genetic characterization, such as the description of dominance/recessiveness, ability to complement existing dpy-23 mutants, description of any dpy-23 like phenotypes, ability to suppress/enhance other dpy-23-mediated processes, etc. Without knowing more about this allele, we cannot reliably interpret the lack of effects seen with this mutant.

We agree that the mew25 allele we use has not been extensively characterized before, however, mew25 and several other alleles introducing the same amino acid change were described previously by other groups (Hollopeter et al., 2014; Beacham et al., 2018; Partlow et al., 2019; Joseph et al., 2020). We have now made this clear in the text and we have included characterization of the allele in Figure 4. We show that no T160 phosphorylation is present in dpy-23(mew25) animals and we show that mew25 rescues the “jowls” phenotype of fcho-1 mutants, likely by destabilizing the closed conformation of AP2 (Hollopeter et al., 2014).

The following text accompanied by new Figure 4 has been added to the Results:

LINE 217 “Characterization of *dpy-23(mew25)* animals revealed that homozygous mutants are viable; they are not dumpy and look superficially wild type. This indicates that the mutants do not suffer from a gross endocytosis defect even though no DPY-23 phosphorylation was detected in *dpy-23(mew25)* animals (Figure 4B). In compliance with previous findings using *dpy-23* alleles harbouring mutations changing the T160 amino acid (Hollopeter et al., 2014, Partlow et al., 2019), *mew25* is able to rescue the “jowls” phenotype of *fcho-1(ox477)* mutants (Figure 4C). FCHO-1 is a member of the muniscin family of proteins and a proposed allosteric activator of AP2 (Hollopeter et al., 2014).”

To further strengthen our findings that the phosphorylation at T160 is not required for the observed QL.d phenotype, we evaluated MIG-14 localization in *dpy-23(mew25)* and we did not observe its re-localization to the plasma membrane. The following sentence accompanied by a figure was added to the Results:

“Furthermore, MIG-14::GFP did not re-localize to the plasma membrane in *dpy-23(mew25)* similar to the *sel-5* mutants and unlike in *dpy-23* RNAi animals (Figure 4E).”

2. I do not agree that, at least from the data shown, "neither levels nor subcellular distribution of a functional MIG-14::GFP protein changed significantly in sel-5 vps-29 compared to vps-29 single mutant". The images shown in Figure 3C are not sufficient to reach this conclusion and I could argue that I see more MIG-14::GFP at the apical and lateral membranes in sel-5 mutants than in control. To make a strong conclusion, the authors need to have higher magnification/resolution images and use markers (e.g. for plasma membrane, junctions, endosomes, golgi, etc.) to carefully analyze the localization of MIG-14::GFP in control vs. mutants.

We agree that the statement about subcellular distribution was not sufficiently supported. We now provide confocal images of MIG-14::GFP expression with higher resolution (Figure 3C) and in Figure 3 —figure supplement 1 we show MIG-14::GFP together with mCherry-PH in egl-20-expressing cells. These data clearly show that MIG-14::GFP does not re-localize to the plasma membrane as would be expected if endocytosis is compromised in sel-5 or sel-5 vps-29 mutants. We use dpy-23 RNAi as a positive control, where MIG-14::GFP co-localization with mCherry-PH membrane marker is visible. The statement cited above was replaced by the following text in Results:

“…levels of MIG-14::GFP did not change significantly in sel-5 vps-29 compared to vps-29 single mutant”

“Importantly, MIG-14::GFP did not re-localize to the plasma membrane in either *sel-5* or *sel-5 vps-29* mutants, in striking contrast to the MIG-14::GFP behaviour in animals treated with *dpy-23* RNAi (Figure 3 —figure supplement 1). This rules out the possibility that the Wnt-related phenotypes in *sel-5 vps-29* mutants arise from a defect in MIG-14 internalization.”

The following text was modified in the Discussion:

“It is possible that either SEL-5 affects endocytosis by a mechanism not dependent on DPY-23 phosphorylation or that SEL-5 regulates QL.d migration and excretory cell canal extension by endocytosis-independent mechanisms. Our findings about MIG-14/Wls behaviour, the most obvious endocytic cargo candidate for QL.d migration, argue against the first possibility; most notably the fact that MIG-14 does not re-localize to the plasma membrane in the absence of SEL-5.”

Similarly, the conclusion that "Levels of MIG-14::GFP in sel-5 single mutants were variable yet on average comparable to wild type" is not supported by Figure 3D, as a statistical analysis of this result is not presented, and it is possible that the high variability makes this result non-significant.

We have added the statistical analysis to Figure 3D, which shows that there is no statistically significant difference in MIG-14 levels between sel-5 and control animals, while vps-29 and sel-5 vps-29 show statistically significant decrease of MIG-14 levels compared to both control and sel-5 animals.

3. I am surprised that the authors reported results from only one multi-copy extrachromosomal array for their studies of SEL-5 kinase mutants. Given the variability in expression from such arrays, typically results are reported for multiple transgenes. More importantly, given that extrachromosomal arrays are known to highly overexpress the protein of interest, and that even proteins with reduced function can rescue when highly overexpressed, the conclusion that SEL-5 kinase activity is not required for function is not supported. These experiments should be done with single-copy insertion transgenes, or by introducing the desired mutations into the endogenous sel-5 locus.

We thank Reviewer 1 for pointing this out. We now include data from more independent lines carrying the mutated SEL-5 constructs (Figure 4 —figure supplement 2) and also from lines carrying the wild type SEL-5 constructs used to assess the tissue-specific action of SEL-5 (Figure 2 —figure supplement 1). We further provide more information about the mutated residues to strengthen our point. The mutated residues are conserved between SEL-5 and AAK1, as shown in Figure 4 —figure supplement 1; when these residues were mutated in AAK1, the kinase activity was lost. Most importantly, the D178A mutation removes the critical aspartic acid of the highly conserved HRD motif, which is considered irreplaceable for catalysis. This makes residual SEL-5D178A kinase activity highly improbable. The following text was added to the Results:

“we repeated the rescue experiment presented in Figure 2A now with SEL-5 carrying either K75A or D178A point mutations. Position D178 corresponds to the D176 of human AAK1 and is part of the conserved HRD motif in the catalytic loop, K75 corresponds to K74 in AAK1 and is predicted to affect ATP binding (Figure 4 —figure supplement 1). In AAK1 both mutations abolished its ability to phosphorylate AP2M1 (Conner and Schmid, 2003).“

In addition, as described for dpy-23(mew25), further genetic characterization of sel-5 kinase mutants is warranted: do they cause phenotypes on their own? Do they affect other sel-5-mediated processes (i.e., look for genetic interactions with lin-12/Notch mutants)? Finally, since the sel-5 coding sequence is tagged with GFP, it would be important to know whether the mutations introduced affect SEL-5 accumulation, localization, etc. Again, this would be better done with either single-copy insertion transgenes or using the endogenous GFP-tagged sel-5.

We thank Reviewer 1 for these important comments. We show at several places in the manuscript (Table 1, Figure 1B, D) that sel-5 single mutants do not display any obvious phenotypes on their own; the only phenotype we have uncovered is a mildly penetrant CAN neuron migration defect (Table 1), but we of course cannot exclude that other less apparent phenotypes might be present in the mutants. The ok149 allele was previously characterized by Fares and Greenwald in 1999 as a loss-of-function allele suppressing the lin-12(d) AC/VU defect. We did not test whether the ok363 allele is able to supress lin-12(d) as Notch signalling was not central to our research. We agree that it would be good to verify the localization of the mutated SEL-5 proteins and that single-copy insertion transgenes or endogenous locus mutations would be better for that than overexpressed protein. However, as we have now observed similar rescue with multiple independent extrachromosomal arrays (Figure 4 —figure supplement 2), we feel our results indicate that the kinase activity of sel-5 is not required for our phenotype.

4. Although the excretory canal outgrowth phenotypes caused sel-5, retromer, and some components of the Wnt signaling, are novel, this story is underdeveloped and lacks mechanistic insight. Moreover, some significant genetic interactions are not considered in the model. Specifically:4a. egl-20/Wnt mutants have a mild, but significant, overgrowth defect, which suggests that EGL-20 acts in parallel to LIN-44 as a "stop" signal (more on this below). This is not mentioned.

The effect of EGL-20 on canal growth is now included in both the Results and Discussion sections together with the novel findings arising from additional mutant combinations – see below.

4b. mig-1/Fzd mutants do not display an excretory canal outgrowth phenotype, however mig-1 loss significantly suppressed the reduced outgrowth caused by loss of cfz-2/Fzd. Moreover, mig-1 did not synergize with egl-20. These results, which are not further discussed, are consistent with a model where egl-20/Wnt and mig-1/Fzd act in a pathway (in parallel to lin-44/Wnt and lin-17/Fzd) as a "stop" signal, and also consistent with a model where even mild disruption of the "stop" signal can overcome defects caused by loss of a "go" signal mediated by cwn-2/Wnt and cfz-2/Fzd. Further genetic interactions (e.g., egl-20 and/or mig-1 mutations with lin-17, which displays a less penetrant phenotype than lin-44) would strengthen this model.

We highly appreciated this feedback as the incorporation of additional Wnt/Frizzled combinations in our analyses brought novel information about Wnt signals involved in excretory canal extension and helped to complement our model. We included several new combinations of Wnt pathway mutants, namely lin-17; egl-20, mig-1 lin-17, cwn-1; cfz-2 and cwn-2; cfz-2 (Figure 6B). Based on the observed phenotypes of these mutants we have modified the text and adjusted our model (Figure 7B) accordingly. Text changes were made in the Abstract, Results and Discussion sections.

The last sentence in the Abstract now states the following: “We further establish that Wnt proteins CWN-1 and CWN-2 together with Frizzled receptor CFZ-2 positively regulate excretory cell outgrowth, while LIN-44/Wnt and LIN-17/Frizzled together generate a stop signal inhibiting its extension.”

The following text is now in the Results section:

“Surprisingly, the loss of egl-20 and mig-1 also partially rescued the canal overgrowth in lin-17 even though neither of them displayed canal shortening on its own, while at the same time mig-1 suppressed canal shortening in cfz-2. Most prominent canal shortening was observed in either cwn-1; cfz-2 or cwn-2; cfz-2 double mutants (Figure 6B). Apart from shortening, three instead of two posterior canal branches were detected in 11% of cwn-1; cfz-2 and 15% of cwn-2; cfz-2 double mutants (Figure 6C).”

The following text about the role of EGL-20 and MIG-1 in canal outgrowth is now included in the Discussion section:

“The role of two other Wnt signalling components, egl-20 and mig-1, is less clear. No effect (mig-1) or only very mild overgrowth defect (egl-20) is observed in single mutants. However, both egl-20 and mig-1 significantly rescue the overgrowth phenotype of lin-17 mutants, while at the same time, mig-1 can suppress the shortening of canals in cfz-2 mutants. EGL-20-producing cells are localized around the rectum (Whangbo et al., 1999; Harterink et al., 2011), exactly where the excretory canals stop, while LIN-44 is expressed more posteriorly (Herman et al., 1995; Harterink et al., 2011). A possible explanation could thus be that while LIN-44 provides a general posterior repulsive signal, EGL-20 fine-tunes the exact stopping position of the growing canal. The role of different Wnts and Frizzleds in excretory canal outgrowth is summarized in Figure 7B.”

5. Implicating EGL-20/Wnt as part of the "stop" signal would allow the authors to use previously-published endogenously-tagged EGL-20 reporters (Pani and Goldstein, 2018) to assess whether the non-autonomous function of sel-5 in QL.d migration and excretory canal outgrowth is related to Wnt secretion, which would provide some mechanistic insight into SEL-5 function. In this regard, I also note that there is a transgenic CWN-2::Venus reporter that could be used to visualize CWN-2/Wnt accumulation in sel-5 mutants (rescue of the cwn-2 excretory canal defect by this transgene should be tested before using).

We agree that an effect on Wnt secretion is a plausible scenario for SEL-5 function. We have therefore generated endogenously-tagged EGL-20::GFP line and assessed EGL-20 gradient formation in control, sel-5, vps-29 and sel-5 vps-29 mutants. Our data, presented in Figure 3 —figure supplement 2, do not support a prominent role of SEL-5 in EGL-20 secretion. It would be interesting to perform the same analysis also for the other Wnts taking part in the excretory cell length regulation, especially CWN-1 and CWN-2. However, the excretory cell phenotypes of sel-5 vps-29 and Wnt mutants are more consistent with a SEL-5/VPS-29 requirement prior to the Wnt-sensitive step, which would argue against a role of SEL-5/VPS-29 in secretion. See also response to Reviewer 2 point 2.

The following text is now in the Results section:

“Our tissue-specific rescue experiments revealed that for a full rescue, simultaneous sel-5 expression is necessary from both egl-20 and hlh-1 promoters. SEL-5 could therefore regulate endocytosis along the EGL-20 transport route, thus shaping the EGL-20 gradient. To test this hypothesis, we endogenously tagged EGL-20 with GFP and assessed Wnt gradient formation in control and mutant backgrounds. This approach has been used previously and revealed that extracellular EGL-20 can be detected in the form of distinct puncta that most likely represent EGL-20 bound to Frizzled (Pani and Goldstein, 2018). We detected EGL-20 puncta in all strains tested, albeit with different frequencies (Figure 3 —figure supplement 2A-C). While the number of EGL-20 puncta in sel-5 mutants was not significantly different from the one in control animals, in sel-5 vps-29 mutants the number of puncta was substantially reduced. However, statistical evaluation did not reveal significant differences between vps-29, sel-5 vps-29 and vps-35; although a decreasing trend in puncta number was visible (Figure 3 —figure supplement 2C). In vps-35, the EGL-20 gradient was shown to be highly reduced or absent (Coudreuse et al., 2006; Harterink et al., 2011b) and was used for comparison. We also compared the signal intensity in the EGL-20 spots among the different strains but no significant differences were detected (Figure 3 —figure supplement 2D). Our data thus do not support the notion that sel-5 significantly affects EGL-20 gradient formation.”

The following text is now in the Discussion section:

“EGL-20 gradient analyses did not detect significant differences between wild type and sel-5 animals, but also not between vps-29 and sel-5 vps-29 in the number of EGL-20 puncta despite the almost 8-fold difference in the QL.d migration phenotype penetrance between these two mutants. This would argue against a SEL-5 role in secretion. However, it is still possible that even a small decrease in secretion efficiency in vps-29 could substantially decrease the frequency of reaching the EGL-20 concentration threshold required to initiate QL migration and such a decrease may not be captured by our assay. A direct correlation between the amount of EGL-20 and the QL.d phenotype would be necessary to answer this question.”

6. Focusing on mig-10 here is confusing. I do not agree that "So far, the most studied pathway regulating excretory canal outgrowth is the one involving UNC-53 […], ABI-1 […], and MIG-10", as many other pathways have been implicated in excretory canal outgrowth and guidance (i.e., Slit/Robo, FGFR, Rho/Rac signaling, the Par polarity pathway, etc.). The authors mention that thus far ligands/receptors that might regulate a UNC-53/ABI-1/MIG-10 are not known, thus perhaps they're suggesting that Wnt signaling would play this role. However, no work is presented to support this model. Thus, as it stands, there is no clear connection between mig-10 and Wnt signaling, and thus the inclusion of mig-10 in this report is not well supported nor necessary.

We agree that understanding the link between MIG-10 and SEL-5 would require more experiments and we have excluded it from the current manuscript.

Reviewer #2 (Recommendations for the authors):1. Please understand that this comment is communicated with the aim of improving the manuscript. I believe the writing quality can be improved with copy-editing. I encourage you to ask a copyeditor to scan the entire document for small errors. Below is one example of the kind of edits needed throughout the manuscript. (My additions are shown in brackets.)From p. 15, line 372:"[A] possible mechanism [for] how SEL-5 could affect endocytosis…"

The text has been copy-edited and we hope it now meets the required standard.

2. In looking up the cited reference for plr-1 mutants, I noticed that the in-text citation misspells the first author's name. (The name is Bhat, rather than Bhatt.) It's possible this error could be caused by a reference manager software, or it is a typo. It would be useful to spell-check all in-text references.

Thank you very much for uncovering this mistake. It has been corrected and all in-text references have been checked.

[Editors’ note: what follows is the authors’ response to the second round of review.]

The manuscript has been improved but there are some remaining issues that need to be addressed, as outlined below:1. The number or intensity of EGL-20 puncta on their own does not provide an accurate description of a morphogenetic gradient. The authors are encouraged to revise the text and/or quantify EGL-20 puncta over distance.2. The evidence implicating sel-5 in only one pathway is not strong. The authors are encouraged to edit the text.3. Both reviewers provide suggestions on how to improve the text. Copy-editing is needed, the EGL-20 gradient should not go under the section titled "SEL-5 does not affect MIG-14 endocytosis". Instead a new section titled "SEL-5 affects the EGL-20 gradient".Reviewer #1 (Recommendations for the authors):The authors have done an excellent job addressing the essential concerns raised in the first round of review with some great new experiments and clarifying changes to the text. In particular, the detailed and complete analysis of excretory canal defects in Wnt pathway mutants allowed the authors to propose an intriguing model describing a novel role for Wnt signaling in canal outgrowth. I also very much appreciate the more detailed analysis of the dpy-23(mew25) allele, which I think lends stronger support to their conclusion that DPY-23/AP2 phosphorylation by SEL-5/AAK1 is not the mechanism by which sel-5 regulates Wnt signaling.However, I do have some concerns on the interpretation of new data re: the EGL-20/Wnt gradient presented in Figure 3, Supplement 2 (In general, I will note that the figures were not labelled in the manuscript; so, it was quite difficult to know what Figure I was looking at when reading the paper). The authors use two criteria to measure the EGL-20 gradient: number of EGL-20 puncta (Figure 3, Supp. 2, C) and signal intensity (Figure 3, Supp. 2, D), and found that neither sel-5 or vps-29 mutations (they do not specify which alleles were used for this experiment, at least I could not find it in the main text, figure legend, methods, or strain table) on their own significantly decreased the number or intensity of particles seen, while the double mutant had significantly fewer particles.

Thank you for pointing out the missing allele, it is the ok149 allele and it is now stated in the Figure legend.

However, I don't think that the number or intensity of EGL-20 puncta on their own provides an accurate description of a gradient. A gradient is not only defined by the total amount of morphogen, but also how the concentration of this morphogen is distributed over distance, and the authors do not include any distance measures in their quantifications. I would argue, based on the representative images shown, that the EGL-20 gradient is indeed quite reduced in sel-5 and vps-29 mutants, because the extent of EGL-20 puncta in these mutants seems to be much shorter than in wildtype. Maybe a good measurement to include is distance from the rectum to the furthest anterior brightest puncta? In conclusion, based on the pictures shown, I agree that sel-5 is likely not significantly affecting the total amount of EGL-20 secreted, but I do not agree that the gradient (which should include a measure of the extent of spread) is unchanged.I also don't think that the EGL-20 gradient should go under the section titled "SEL-5 does not affect MIG-14 endocytosis". Instead a new section titled "SEL-5 affects the EGL-20 gradient" (or "does not affect" depending on how the new measurements of gradient that incorporate distance turn out) should be added, and these figures should be part of the main text.

We have performed the measurements of the length of the EGL-20 gradient as suggested and indeed, the gradient is shorter in the sel-5 vps-29 double mutants. However, there is no statistically significant difference between gradient length in vps-29 and sel-5 vps-29 or vps-35 mutants despite the big difference in the migration phenotype. This is mostly because the gradient lenght in vps-29 is highly variable. It seems plausible that in vps-29 the Wnt secretion is variable and therefore, with some frequency, there is enough EGL-20 reaching to the Q neuroblast and the Wnt pathway is activated, while in the sel-5 vps-29 double mutants the variability in EGL-20 secretion adds-up with a slight decrease in EGL-20 spreading efficiency caused by sel-5 and the gradient does not reach far enough to activate the pathway. However, since at the moment we are not able to directly correlate the length of the gradient and the migration phenotype and since we do not know the mechanism explaining how SEL-5 could reduce EGL-20 spreading, we do not want to over-interpret the phenotype.

The following changes have been made:

Original Figure 3 —figure supplement 2 is now Figure 4 and it has been moved to the main text, the new data are part of the Figure 4 as panel 4E. All subsequent Figures have been re-numbered. The text is now moved to a separate section entitled “SEL-5 helps to shape the EGL-20 gradient”.

The following text changes have been made:

Results section:

“SEL-5 helps to shape the EGL-20 gradient

Our tissue-specific rescue experiments revealed that for a full rescue, simultaneous sel-5 expression is necessary from both egl-20 and hlh-1 promoters. SEL-5 could therefore regulate endocytosis along the EGL-20 transport route, thus shaping the EGL-20 gradient. To test this hypothesis, we endogenously tagged EGL-20 with GFP and assessed Wnt gradient formation in control and mutant backgrounds. This approach has been previously used and revealed that extracellular EGL-20 can be detected in the form of distinct puncta that most likely represent EGL-20 bound to Frizzled (Pani and Goldstein, 2018). We detected EGL-20 puncta in all strains tested, albeit with different frequencies (Figure 4A-C). While the number of EGL-20 puncta in sel-5 mutants was not significantly different from that in control animals, in sel-5 vps-29 mutants the number of puncta was substantially reduced. However, statistical evaluation did not reveal significant differences between vps-29, sel-5 vps-29 and vps-35; although a decreasing trend in puncta number was visible (Figure 4C). In vps-35, the EGL-20 gradient was shown to be highly reduced or absent (Coudreuse et al., 2006; Harterink et al., 2011b) and was used for comparison. We also compared the signal intensity in the EGL-20 spots among the different strains but no significant differences were detected (Figure 4D). Finally, we assessed the length of the gradient by measuring how far from the rectum the EGL-20 puncta can be detected. While EGL-20 gradient formation in neither sel-5 nor vps-29 was significantly affected, the EGL-20 gradient was shorter in vps-29 sel-5 double mutants than in wild type controls (Figure 4E). Our data thus support the notion that sel-5 plays a subtle role in EGL-20 gradient formation.”

Discussion section:

“However, the effect of SEL-5 on a different endocytic cargo cannot be excluded. We detected significant shortening of the EGL-20 gradient in the sel-5 vps-29 mutants. Given that SEL-5 is required both in the EGL-20-producing cells and in the muscle for QL.d migration, it is possible that some receptors that serve as a sink for EGL-20 are incorrectly accumulated at the plasma membrane and thus partially hinder EGL-20 spreading. In case of the excretory canal extension, SEL-5 could regulate the turnover of guidance receptors at the plasma membrane of the growing tip.”

“SEL-5 could thus affect Wnt secretion in the case of QL.d migration or new guidance receptor delivery to the plasma membrane in the case of the excretory canals. EGL-20 gradient analyses revealed gradient shortening in sel-5 vps-29 mutants. However, no significant differences in the number of EGL-20 puncta were detected between wild type and sel-5 animals, but also not between vps-29 and sel-5 vps-29, despite the almost 8-fold difference in the QL.d migration phenotype penetrance between these two mutants. This would argue against a role for SEL-5 in secretion. Shorter gradient in the double mutants could thus more likely arise from a combination of partially reduced EGL-20 secretion caused by vps-29 and a partial reduction of EGL-20 spreading caused by a possible sel-5-dependent membrane retention of a receptor acting as an EGL-20 sink. A direct correlation between the amount of EGL-20, the range of the EGL-20 gradient and the QL.d phenotype would be necessary to explain the phenotypic differences between the various mutants. More experiments will thus be required to establish how SEL-5 exerts its function.”

The Figure legend has been updated: (“E) The length of the EGL-20 gradient was assessed in the same images as in (C). The distance from the rectum to the most distant clearly recognizable EGL-20 particle was measured in blinded images. Measurement was repeated three times; the values were averaged and plotted as a single data point. For (C, D and E) Wilcoxon rank sum test was performed to assess statistical significance. Bonferroni correction for multiple testing was applied. * p-value < 0.005,** p-value < 0.001, error bars represent 95% confidence interval, only statistically significant differences shown in (C and E), no statistically significant differences revealed in (D)”

Reviewer #2 (Recommendations for the authors):Overall, I was satisfied by most changes in the revised manuscript.Copy-editing was needed. There were a large number of syntax errors, particularly missing articles, prepositions, and punctuation. This was true in the introduction, results, and discussion.

We have utilized the free Grammarly Grammar Checker platform (www.grammarly.com) and consulted the text with a native speaker and we now hope the text meets the expected standard.